# LoGra-Med: Long Context Multi-Graph Alignment for Medical Vision-Language Model

## Abstract

State-of-the-art medical multi-modal large language models (med-MLLM), such as LLAVA-MED or BIOMEDGPT, leverage instruction-following data in their pre-training stages. However, those models primarily focus on scaling the *model size* and *data volume* to boost performance while mainly relying on the autoregressive learning objectives. Surprisingly, we reveal that such learning schemes might result in a weak alignment between vision and language modalities, making these models highly reliant on extensive pre-training datasets — a significant challenge in medical domains due to the expensive and time-consuming nature of curating high-quality instruction-following instances. We address this challenge with a new multi-graph alignment algorithm, namely LoGra-Med, which enforces triplet correlations on the latent embedding space among image modalities, conversation-based descriptions, and extended contextual captions. Owing to this technique, the model is encouraged to capture the semantic meaning of the context, handle linguistic variability where the captions or questions may differ from training instances, and learn cross-modal associations, linking visual elements with various textual interpretations. To scale our algorithm to the med-MLLM setting, we also design an efficient end-to-end learning scheme based on advanced black-box gradient-estimation techniques that permit fast forward and backward steps through the LLM model (LLaMa 7B). Empirical results show that we can match the performance of LLAVA-Med pre-trained on 600K image-text pairs from PMC-15M for Medical VQA tasks and significantly outperform it when trained on only $10\%$ of the data. For instance, on VQA-RAD, we exceed LLAVA-Med (both trained on $10\%$) by $20.13\%$ and achieve near parity with the $100\%$ pre-training setting ($72.52\%$ vs. $72.64\%$). Additionally, we also surpass other SOTA pre-training methods and med-MLLM such as BIOMEDGPT on *visual chatbot* or RADFM on *zero-shot image classification with VQA*, showcasing the power of multi-graph alignment in improving vision-language integration for medical-MLLM.

## 1 Introduction

Generic Multi-Modal Large Language Models (MLLM) are an emerging field integrating processing and generation across text, images, and audio. Models like GPT-4V (Achiam et al., 2023), LLaVa (Liu et al., 2024b), and Next-GPT (Wu et al., 2023b) handle tasks from image captioning to complex visual reasoning. A critical component in training these models is instruction-following (IF) data (Lou et al., 2023), which involves complex, multi-turn interactions (Sun et al., 2024) where the model is expected to respond to specific instructions or questions about the image. In the medical domain, current efforts have been focused on building medical MLLM by curating specialized IF datasets encompassing medical images, clinical notes, and diagnostic criteria (Xie et al., 2024). These datasets are used to adapt general-purpose MLLM, aiming to transfer foundational knowledge of generic concepts and reduce computation costs required in training steps. For instance, in LLAVA-Med, (Li et al., 2024) 600K biomedical image-text pairs are sampled from PMC-15M (Zhang et al., 2023c) and GPT-4 is used to create instruction data from the text inputs, resulting in approximately 60K multi-modal IF data points. In the next step, (i) vision encoders and language decoders are taken from LLaVa and are first aligned through trainable projection layers before (ii) the models (with the exception of the vision encoders) are trained together on collected medical IF data. Both steps are called the *pre-training* phase, where the auto-regressive function is used as the primary objective. The model can then be fine-tuned to various downstream tasks.

Following the above approach, most later works have focused on scaling up the amount of medical IF data (Xie et al., 2024; Zhang et al., 2023a; He et al., 2024) or increasing the model size by incorporating larger vision encoders or language decoders (Wu et al., 2023a; Jiang et al., 2024) while relying on the same standard autoregressive learning scheme. Contrary to this, we question the effectiveness of autoregressive objective functions when learning medical-MLLM with IF data. Surprisingly, *our findings reveal that autoregressive learning is highly data-hungry during pre-training*, i.e., without sufficient medical IF samples, model performance plummets for downstream tasks, *even after fine-tuning*. To illustrate this, we pre-trained LLAVA-Med using only 10% of the

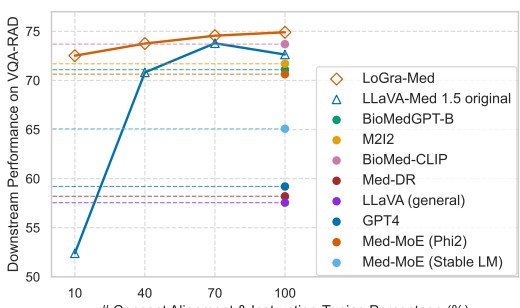

Figure 1: *Illustration of the data-hungry behavior of auto-regressive modeling* in LLaVA-Med when varying pre-training IF data size. Models are fine-tuned and performance is reported on VQA-RAD.

data and compared it to the version trained on 100%. Both models were fine-tuned on two medical visual question-answering tasks - VQA-RAD (Lau et al., 2018) and PathVQA (He et al., 2020) - and their average performance on open- and close-ended questions compared. The results show a dramatic decline: from 72.64% to 52.39% on VQA-RAD and from 64.06% to 56.15% on PathVQA (Figure 1). This underscores the instability of medical-MLLM trained with autoregressive methods and highlights the problem that these methods require the curation of enough medical IF data to achieve satisfactory performance.

To address this challenge, we present a novel multi-graph alignment algorithm, namely LOGRA-MED, that improves the model's ability to learn complex interactions between vision and language modalities, mitigating the limitations of autoregressive functions when trained on limited instruction-following data. Specifically, given pairs of input images with instruction data, we use GPT-4 (Achiam et al., 2023) to *form a longer version of the instruction*, with more in-context explanations for concepts and correlations among entities while preserving the same meaning. These data pairs are fed into the MLLM, where a vision encoder extracts features for the image while the language model (LLaMa) computes latent embedding of the instruction data and its extended versions. The embedding vectors obtained from different IF samples in a batch are then used to construct three graphs. The first graph's vertices are the image features, while the vertices of the two other graphs are the embeddings of the instructions and their extended versions. We subsequently learn feature representation by solving the combinatorial multi-graph alignment problem between these three graphs such that the input embedding of the graph alignment will output the triplet coupling among the image, its instruction data, and the longer extension. By jointly learning this alignment with autoregressive methods, we aim to preserve the LLMs' sequential generation capabilities while enhancing the model's ability to capture deeper semantic relationships. This approach goes beyond memorization, increases resilience to linguistic variability, and significantly improves cross-modal coherence.

Our method differentiates itself from existing multi-modal alignment techniques for LLM (Park et al., 2024; Li et al., 2023a; Chen et al., 2023a) in two ways. First, while prior contrastive objectives primarily focus on *learning projections* to connect frozen vision encoders with frozen language models, our algorithm directly trains LLM using the multi-graph framework. Second, we unify and generalize pairwise contrastive learning, which focuses on individual image-caption pairs, by incorporating global constraints through graph structures. This allows us to integrate both feature and structural consistencies using graph edges, enhancing robustness for similar entities (whether images or descriptions) commonly found in medical datasets. Although combinatorial graph alignment is generally non-differentiable and solving multi-graph alignment is computationally expensive, we overcome these challenges by leveraging modern implicit maximum likelihood estimation techniques (Niepert et al., 2021; Minervini et al., 2023). This enables efficient gradient estimation and allows for fast forward and backward propagation through large LLMs (e.g., LLaMa-7B), using a *barycenter graph* (Agueh & Carlier, 2011) for alignment. As a result, the model can scale effectively with extensive datasets on large LLMs while maintaining alignment performance.

In summary, we make the following key contributions:

- We reveal the data-demanding nature of autoregressive modeling in pre-training medical-MLLM (LLaVa-Med), showing that insufficient instruction-following data leads to significant performance drops on downstream tasks, even after fine-tuning.

- We introduce a novel multi-graph alignment objective that establishes triplet constraints among images, their instruction-following context, and their enriched versions. Additionally, we developed an efficient solver for training with LLMs and outlined theoretical properties related to distance and the shortest path in the geodesic space of multi-modal graphs.

- We demonstrate that using a small amount of pre-training data can achieve performance comparable to LLaVa-Med trained on $100\%$ data. Additionally, when trained on larger datasets, LOGRA-MED outperforms several state-of-the-art medical mLLMs and advanced multi-modal pre-training algorithms across three Medical VQA tasks, medical visual chat, and the average zero-shot image classification performance on 23 datasets.

## 2 RELATED WORK

**Medical Multi-modal LLMs.** Recent developments in medical-MLLM like Biomed-GPT (Zhang et al., 2023a), MedFlamingo (Moor et al., 2023), Med-Dr (He et al., 2024), LLAVA-Med (Li et al., 2024), and Med-PaLMs (Singhal et al., 2023; Tu et al., 2024) are transforming healthcare by integrating diverse data types and scaling medical instruction data. Biomed-GPT excels with multiple biomedical modalities, MedFlamingo focuses on few-shot learning for medical visual question answering, and LLAVA-Med leverages large-scale biomedical image-text pairs for improved performance. Commonly, these models emphasize scaling medical instruction data and increasing model parameters to enhance accuracy and applicability in real-world medical scenarios. In contrast, our approach examines the widely used autoregressive pre-training algorithms and demonstrates that incorporating long-context multi-graph alignment of exist instruction samples can significantly enhance medical-MLLM performance without requiring larger models or extensive datasets.

**Visual Instruction Tuning.** Visual instruction tuning techniques aim to bridge the gap between frozen vision-language models and frozen LLMs trained on unimodal data, enabling them to work effectively in a multi-modal context. These methods involve (i) learning a multi-layer perceptron (MLP) layer to map embeddings from the vision model to the language model as LLaVa (Liu et al., 2024b), VideoLLM (Chen et al., 2023a); (ii) using adapter-based adjustment as LLaMa-adapter (Zhang et al., 2024), Voxposer (Huang et al., 2023), or (iii) learning multi-modal perceiver by gated cross-attention (Alayrac et al., 2022) or Q-Former as in BLIP-2 (Li et al., 2023a). Pre-training algorithms to train these models can be combined with both auto-regressive and contrastive learning (Park et al., 2024; Zhai et al., 2023) or image-text matching as in (Li et al., 2022; 2023a). Our algorithm differs from those by focusing on directly training LLMs rather than lightweight projectors. This requires a fast solver capable of efficiently handling forward and backward passes through large-scale LLMs with extensive parameters.

**Vision-language Pretraining Algorithm.** Pre-training algorithms commonly applied for vision-language models, like CLIP (Radford et al., 2021), follow various strategies. Generative approaches, such as masked prediction in language models (Devlin, 2018; Song et al., 2020), or autoregressive algorithms that predict sequential text in LLMs (Liu et al., 2024b; Zhang et al., 2024), are widely used. Another direction focuses on discriminative methods, which learn contrastive distances between image-text pairs (Liu et al., 2023; Zhai et al., 2023; Khan & Fu, 2023), optimal transport (Chen et al., 2022; Nguyen et al., 2024a), or impose clustering constraints (Park et al., 2024). Our function departs from these by generalizing them into a combinatorial graph-matching formulation across cross-domain graphs. While LVM-Med (MH Nguyen et al., 2024) is the most similar to our approach, it targets alignment within vision tasks, whereas we align images, instruction-following data, and extended contextual information.

Graph alignment across $K$ domains ($K \geq 3$) is highly computationally intensive. Current methods, such as multi-marginal optimal transport (Lin et al., 2022; Piran et al., 2024), Wasserstein barycenters (Nguyen et al., 2024b), and multi-adjacency matrix assumptions (Bernard et al., 2019; Swoboda et al., 2019), relax the problem but are limited to small-scale tasks and require multiple solver steps, making them inefficient for LLM training. In contrast, our approach utilizes heuristic solvers (Swoboda et al., 2017; Rolínek et al., 2020) and modern gradient estimation techniques for black-box optimization (Niepert et al., 2021; Minervini et al., 2023), enabling scalable and efficient performance for large language models.

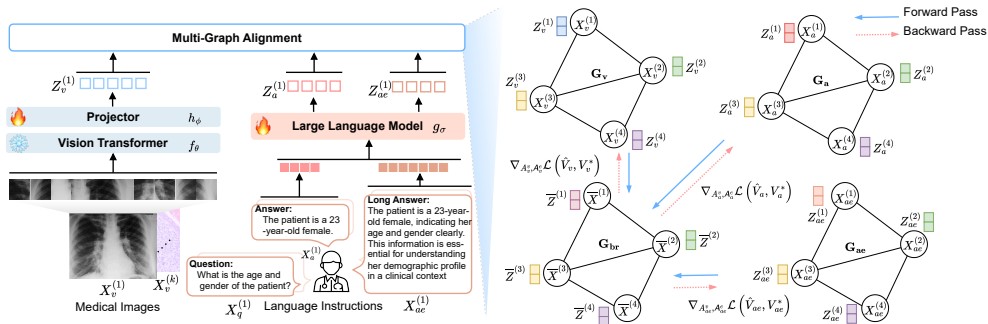

Figure 2: Overview of LOGRA-MED: We train the large language model $g_\sigma$ and projector $h_\phi$ by aligning a triplet consisting of the input image, its instruction data, and its extended long-context version using a structure-aware multi-graph alignment between $\mathcal{G}_v$, $\mathcal{G}_a$, and $\mathcal{G}_{ae}$ through a barycenter graph $\mathcal{G}_{br}$.

## 3 MULTI-GRAPH ALIGNMENT LEARNING

We denote the vision encoder, projector, and large-language model (LLM) models are $f_\theta(.)$, $h_\phi(.)$, $g_\sigma(.)$, respectively. Figure 2 illustrates our LOGRA-MED algorithm, which learns parameters for these models by solving a triplet alignment between modalities in instruction tuning data. Below, we summarize the notations used before describing each component in detail.

**Notation.** Given any tensor $\mathsf{T} = (T_{i,j,k,l})$ and matrix $\boldsymbol{M} = (M_{k,l})$, we use $\mathsf{T} \otimes \boldsymbol{M}$ to denote the tensor-matrix multiplication, *i.e.,* the matrix $(\sum_{k,l} T_{i,j,k,l} M_{k,l})_{i,j}$. Given $\boldsymbol{Y} = [\boldsymbol{y}_1, \boldsymbol{y}_2, ..., \boldsymbol{y}_N] \in \mathbb{R}^{N \times d}$, we define $\mathbb{E}(\boldsymbol{Y}) = \frac{1}{N} \sum_{i=1}^{N} \boldsymbol{y}_i \in \mathbb{R}^d$. Moreover, we define the matrix scalar (or inner) product associated with the Frobenius norm between two matrices $\boldsymbol{M} = (M_{i,j})$ and $\boldsymbol{N} = (N_{i,j})$ as $\langle \cdot, \cdot \rangle$, *i.e.,* $\langle \boldsymbol{M}, \boldsymbol{N} \rangle = \sum_{i,j} M_{i,j} N_{i,j}$. We write $[M] = \{1, 2, \ldots, M\}$ for any natural number $M$.

### 3.1 LONG-CONTEXT ENRICHED MEDICAL INSTRUCTION-FOLLOWING DATA

Recent research has demonstrated that incorporating long-context significantly enhances LLMs' ability to process complex inputs and improves instruction-following by retaining more relevant information (Liu et al., 2024c; An et al., 2024; Pawar et al., 2024). Building on this insight, we extend medical instruction-following data by generating *long-context paraphrased versions of existing samples in the dataset*, providing an additional enriched perspective of the original data.

In particular, a typical instruction sample includes $\{\boldsymbol{X}_v, [\boldsymbol{X}_q^1, \boldsymbol{X}_a^1], ..., [\boldsymbol{X}_q^L, \boldsymbol{X}_a^L]\}$ where $\boldsymbol{X}_v$ is an input image, $\boldsymbol{X}_q^l$ a question, and $\boldsymbol{X}_a^l$ an answer at round $l$ in multi-round $L$ of a conversation. In the medical domain, most of the questions are generic, and the information answer usually covers the question, so we only focus on extending the answer $\boldsymbol{X}_a$. We leverage the GPT API with a `prompt` to form a longer-context for each $\boldsymbol{X}_a^l$ by:

$$\boldsymbol{X}_{ae}^l = \text{GPT}\left(\boldsymbol{X}_q^l, \boldsymbol{X}_a^l, \text{prompt}\right), \forall l \in [L]. \tag{1}$$

The details for `prompt` are presented in the Appendix. In short, we ask GPT to provide additional explanations for concepts that appeared in the original answer $\boldsymbol{X}_a$ while keeping the content consistent. An example output for $\boldsymbol{X}_{ae}^l$ is illustrated in Figure 3.

### 3.2 MULTI-GRAPH CONSTRUCTION ON VISION-LANGUAGE EMBEDDING

For each *image* $\boldsymbol{X}_v \in \mathbb{R}^{3 \times H \times W}$ where $(H, W)$ are the original spatial dimensions. We divide it into a sequence of visual patches $\boldsymbol{U} = [u_i]_{i=1}^N$ with $N = (H \times W)/U$ with $U$ as the patch size. Using a pre-trained ViT model $f_\theta$, we extract patch-wise features as $\boldsymbol{V} = f_\theta(\boldsymbol{U}) \in \mathbb{R}^{N \times d_v}$ and apply another projector to map it into the projected embedding $\boldsymbol{Z} = h_\phi(\boldsymbol{V}) \in \mathbb{R}^{N \times d}$. We then pool the features from the image patches to define a global description as $\boldsymbol{Z}_v = \mathbb{E}(\boldsymbol{Z}) \in \mathbb{R}^d$. For each *language input* $\boldsymbol{X}_c^l \in \{\boldsymbol{X}_a^l, \boldsymbol{X}_{ae}^l\}$ with $c \in \{a, ae\}$, we assume it has $M$ tokens, i.e., $\boldsymbol{X}_c^l = [\boldsymbol{x}_j]_{j=1}^M \in \mathbb{R}^M$, and feed it into the LLM model to extract a set of embedding $\boldsymbol{Z}_c^l = g_\sigma([\boldsymbol{x}_j]_{j=1}^M) = [\boldsymbol{e}_j]_{j=1}^M \in \mathbb{R}^{M \times d}$. We subsequently concatenate all multi-round $L$ in each single instruction tuning to define $\boldsymbol{Z}_c = \frac{1}{L} \sum_{l=1}^{L} \mathbb{E}(\boldsymbol{Z}_c^l)$ which collects average text embedding of original answers ($c = a$) and their longer-context extended versions ($c = ae$) respectively.

Given a batch size of $B$ instruction-tuning samples, we now construct three graphs $\mathcal{G}_v = (\mathcal{V}_v, \mathcal{E}_v)$, $\mathcal{G}_a = (\mathcal{V}_a, \mathcal{E}_a)$, and $\mathcal{G}_{ae} = (\mathcal{V}_{ae}, \mathcal{E}_{ae})$ representing for visual image features, text embedding encoded by LLM for original answers and their long-context embedding extended by GPT. Specifically, for each triplet pair $\{\boldsymbol{X}_v^{(k)}, [\boldsymbol{X}_a^l]^{(k)}, [\boldsymbol{X}_{ae}^l]^{(k)}\}_k$, $(k \in [B])$, we add a node representing $\boldsymbol{X}_v^{(k)}$ to $\mathcal{V}_v$, a node for $[\boldsymbol{X}_a^l]^{(k)}$ to $\mathcal{V}_e$, and finally a node for $[\boldsymbol{X}_{ae}^l]^{(k)}$ to $\mathcal{V}_{ae}$. This results in a set of nodes $\mathcal{V}_v = \{\boldsymbol{X}_v^{(1)}, ..., \boldsymbol{X}_v^{(B)}\}$; $\mathcal{V}_c = \{[\boldsymbol{X}_c^l]^{(1)}, ..., [\boldsymbol{X}_c^l]^{(B)}\}$ for each $c \in \{a, ae\}$. We equip node-level feature matrices for these graphs using their embedding computed above, i.e., $\boldsymbol{F}_v = \{\boldsymbol{Z}_v^{(1)}, ..., \boldsymbol{Z}_v^{(B)}\}$, $\boldsymbol{F}_c = \{\boldsymbol{Z}_c^{(1)}, ..., \boldsymbol{Z}_c^{(B)}\}$. The edges for $\mathcal{E}_v, \mathcal{E}_c$ afterward can be created through the k-nearest neighbors algorithm

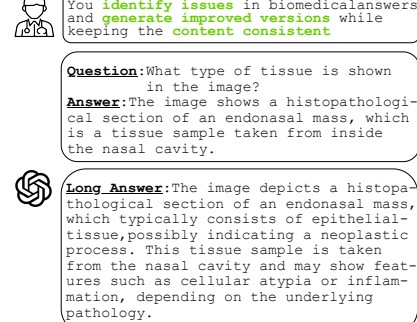

Figure 3: Illustration for creating the longer-context instruction-following data powered by GPT-4o.

given the feature node matrices $\boldsymbol{F}_v$, $\boldsymbol{F}_c$. Finally, we can run a message-passing network $m_\alpha(.)$ on three built graphs to learn richer node representations. This approach has proven effective for representation learning (Tang et al., 2022; Ju et al., 2024), resulting in aggregated feature-node matrices as $\{\hat{\boldsymbol{Z}}_s^{(1)}, ..., \hat{\boldsymbol{Z}}_s^{(B)}\} = m_\alpha(\boldsymbol{F}_s, \mathcal{E}_s)$, with $s \in \{v, a, ae\}$.

### 3.3 SECOND-ORDER GRAPH ALIGNMENT PROBLEM

We first provide background about the second-order graph alignment between two arbitrary graphs $\mathcal{G}_1 = (\mathcal{V}_1, \mathcal{E}_1)$ and $\mathcal{G}_2 = (\mathcal{V}_2, \mathcal{E}_2)$, which is mentioned as quadratic assignment problem and applied in several problems in vision and graphics to find correspondences between two graph structures under *consistent between node features and structure constraints* (Zanfir & Sminchisescu, 2018; Haller et al., 2022; Ehm et al., 2024).

We denote by $\boldsymbol{V} \in \{0, 1\}^{|\mathcal{V}_1||\mathcal{V}_2|}$, with $|\mathcal{V}_1| = M$ and $|\mathcal{V}_2| = N$, the indicator matrix of matched vertices, that is, $V_{i,j} = 1$ if a vertex $v_i \in \mathcal{V}_1$ is matched with $v_j \in \mathcal{V}_2$ and $V_{i,j} = 0$ otherwise. That is, $\boldsymbol{V}$ is a binary matrix with exactly one non-zero entry in each row and column. Similarly, we set $\mathsf{E} \in \{0, 1\}^{|\mathcal{E}_1||\mathcal{E}_2|}$ as the indicator tensor of match edges, that is, $E_{i,k,j,l} = 1$ if $V_{i,j} = 1$ and $V_{k,l} = 1$ and $E_{k,l} = 0$ otherwise. This implies that the tensor $\mathsf{E}$ is fully determined by the matrix $\boldsymbol{V}$, that is, $E_{i,k,j,l} = V_{i,j}V_{k,l}$. We also define the vertex affinity matrix and edge affinity tensor as $\boldsymbol{A}^v \in \mathbb{R}^{|\mathcal{V}_1||\mathcal{V}_2|}$ and $\mathsf{A}^e \in \mathbb{R}^{|\mathcal{E}_1||\mathcal{E}_2|}$, respectively. The set $\mathcal{A}(\mathcal{G}_1, \mathcal{G}_2)$ indicates for all admissible pairs $(\boldsymbol{V}, \mathsf{E})$ that encode a valid matching between $\mathcal{G}_1$ and $\mathcal{G}_2$.

$$\mathcal{A}(\mathcal{G}_1, \mathcal{G}_2) = \left\{ \boldsymbol{V} \in \{0, 1\}^{M \times N} : \sum_{i=1}^{M} V_{i,j} = 1, \sum_{j=1}^{N} V_{i,j} = 1 \right\}. \tag{2}$$

The second-order graph alignment (SoGA) problem now is defined as:

$$\text{SoGA}(\boldsymbol{A}^v, \mathsf{A}^e) = \underset{\boldsymbol{V} \in \mathcal{A}(\mathcal{G}_1, \mathcal{G}_2)}{\arg\min} \langle \boldsymbol{A}^v + \mathsf{A}^e \otimes \boldsymbol{V}, \boldsymbol{V} \rangle = \underset{\boldsymbol{V} \in \mathcal{A}(\mathcal{G}_1, \mathcal{G}_2)}{\arg\min} \sum_{i,j} A_{i,j}^v V_{i,j} + \sum_{i,j,k,l} \mathsf{A}_{i,j,k,l}^e V_{i,j} V_{k,l}. \tag{3}$$

### 3.4 SCALABLE MULTI-GRAPH ALIGNMENT

Our aim is to solve the graph alignment between three graphs $\mathcal{G}_v$, $\mathcal{G}_a$, and $\mathcal{G}_{ae}$ to form a triplet constraint between input image embedding, its original instruction embedding, and the long-context extended version ones. However, solving a structure-aware graph alignment between $K$ domains is computationally expensive. One potential solution is to perform pairwise graph alignments $\binom{K}{2}$ times, as shown in Eq. (3), while applying specific constraints to maintain consistency between correspondences (Bernard et al., 2019; Swoboda et al., 2019). However, this method becomes impractical as $K$ increases, making it unsuitable for larger-scale problems.

Another direction leverages the barycenter concept from optimal transport, which identifies a central distribution that minimizes the weighted sum of Wasserstein distances to the given input distributions. We follow this idea to reformulate the alignment of $K$ graphs into $K$ separate alignments with a barycenter graph. Unlike previous unsupervised methods that estimate the barycenter before aligning, we directly define the barycenter using known triplet pairs across the three graphs. This significantly reduces complexity, making our solver more efficient in LLM settings.

Specifically, we define a new barycenter graph $\mathcal{G}_{br} = (\mathcal{V}_{br}, \mathcal{E}_{br})$ where $\mathcal{V}_{br} = \left\{ v_{br}^{(1)}, ..., v_{br}^{(B)} \right\}$ with $v_{br}^{(k)} = \overline{\boldsymbol{X}}^{(k)} = \left\{ \boldsymbol{X}_v^{(k)}, [\boldsymbol{X}_a^l]^{(k)}, [\boldsymbol{X}_{ae}^l]^{(k)} \right\}$ and a correspondence feature node as $\boldsymbol{F}_{br} = \frac{1}{3} \left\{ \sum_s \hat{\boldsymbol{Z}}_s^{(1)}, ..., \sum_s \hat{\boldsymbol{Z}}_s^{(B)} \right\}$ with $s \in \{v, a, ae\}$. The edge set $\mathcal{E}_{br}$ is formed similarly to another graph by running the k-nearest neighbor on feature node $\boldsymbol{F}_{br}$. We now state the multi-graph alignment as:

$$\text{SoGA}(\boldsymbol{A}_s^v, \mathbf{A}_s^e) = \underset{\boldsymbol{V}_s \in \mathcal{A}(\mathcal{G}_s, \mathcal{G}_{br})}{\arg\min} \sum_{s \in \{v, a, ae\}} \langle \boldsymbol{A}_s^v + \mathbf{A}_s^e \otimes \boldsymbol{V}_s, \boldsymbol{V}_s \rangle, \tag{4}$$

where $\boldsymbol{V}_s$ is the indicator matrix representing for valid mapping between $\mathcal{G}_s$ and $\mathcal{G}_{br}$, $\boldsymbol{A}_s^v \in \mathbb{R}^{|\mathcal{V}_s||\mathcal{V}_{br}|}$ and $\mathbf{A}_s^e \in \mathbb{R}^{|\mathcal{E}_s||\mathcal{E}_{br}|}$ be vertex affinity matrix and edge affinity tensor between $\mathcal{G}_s$ and $\mathcal{G}_{br}$. For e.g., $(\boldsymbol{A}_s^v)_{ij} = d\left( \hat{\boldsymbol{Z}}_s^{(i)}, \frac{1}{3} \sum_s \hat{\boldsymbol{Z}}_s^{(j)} \right)$ with $d(.)$ be a distance metric (e.g., cosine distance) measuring similarity between node $i^{th}$ in $\mathcal{G}_s$ and node $j^{th}$ in $\mathcal{G}_{br}$.

To address the NP-Hard nature of aligning each graph to the barycenter graph $\mathcal{G}_c$, which arises from its combinatorial complexity, we employ efficient heuristic solvers utilizing Lagrange decomposition techniques (Swoboda et al., 2017; Rolínek et al., 2020).

### 3.5 Backpropagation with Black-box Gradient Estimation

Given $\hat{\boldsymbol{V}}_s = \text{SoGA}(\boldsymbol{A}_s^v, \mathbf{A}_s^e)$ be solution obtained from the solver, we aim to learn feature representation for LLMs such that $\hat{\boldsymbol{V}}_s$ be identical to true triplet alignments explicitly indicated by the barycenter graph. By denoting $\boldsymbol{V}_s^*$ be an optimal mapping between the graph $\mathcal{G}_c$ to $\mathcal{G}_{br}$, we compute the following total of hamming loss function:

$$\mathcal{L}(\hat{\boldsymbol{V}}_s, \boldsymbol{V}_s^*) = \sum_{s \in \{v, a, ae\}} \langle \hat{\boldsymbol{V}}_s, (1 - \boldsymbol{V}_s^*) \rangle + \langle \boldsymbol{V}_s^*, (1 - \hat{\boldsymbol{V}}_s) \rangle. \tag{5}$$

However, computing the gradient of the loss function with respect to the matching problem inputs $(\boldsymbol{A}_s^v, \mathbf{A}_s^e)$, i.e., $\nabla_{\boldsymbol{A}_s^v, \mathbf{A}_s^e} \mathcal{L}\left( \hat{\boldsymbol{V}}_s, \boldsymbol{V}_s^* \right)$, poses a challenge due to the piecewise constant nature of the graph matching objective in Eq. (4) (Pogančić et al., 2020; Rolínek et al., 2020). To address this, we resort to the IMLE techniques (Niepert et al., 2021; Minervini et al., 2023), a method permitting estimate gradients over solutions of the combinatorial optimization problem by taking the difference between solutions of matching problem perpetuated by Gumbel noise.

In particular, given $(\epsilon, \epsilon') \sim \text{Gumble}(0, 1)$ and for each $s \in \{v, a, ae\}$, we compute:

$$\left( \frac{\partial \mathcal{L}}{\partial \boldsymbol{A}_s^v}, \frac{\partial \mathcal{L}}{\partial \mathbf{A}_s^e} \right) \approx \tilde{\boldsymbol{V}}_s - \text{SoGA}\left( \boldsymbol{A}_{s,\lambda}^v, \mathbf{A}_{s,\lambda}^e \right) \text{ where } \tilde{\boldsymbol{V}}_s = \text{SoGA}\left( \boldsymbol{A}_s^v + \epsilon, \mathbf{A}_s^e + \epsilon' \right), \tag{6}$$

$$\left( \boldsymbol{A}_{s,\lambda}^v, \mathbf{A}_{s,\lambda}^e \right) = \left( \boldsymbol{A}_s^v + \epsilon, \mathbf{A}_s^e + \epsilon' \right) - \lambda \nabla_{\tilde{\boldsymbol{V}}_s} \mathcal{L}(\tilde{\boldsymbol{V}}_s, \boldsymbol{V}_s^*), \text{ with } \lambda \text{ is a step size.} \tag{7}$$

### 3.6 Structure alignment-based graph distance properties

In this section, we present theoretical insights into the graph-matching problem outlined in Eq. (3). Specifically, we demonstrate that once the optimal matching between two graphs is established, it defines a valid metric distance. Additionally, the geodesic path (i.e., the shortest path distance) connecting the two graphs in the manifold space can be derived based on the computed matching alignments. We define a discrete between two graphs given a solution of matching alignment as:

$$d_{\text{SoGA}}(\mathcal{G}_1, \mathcal{G}_2) = \min_{\boldsymbol{V} \in \mathcal{A}(\mathcal{G}_1, \mathcal{G}_2)} \left( \sum_{i,j} A_{i,j}^v V_{i,j} + \sum_{i,j,k,l} \boldsymbol{A}_{i,j,k,l}^e V_{i,j} V_{k,l} \right). \tag{8}$$

We consider labelled graphs as tuples of the form $\mathcal{G} = (\mathcal{V}, \mathcal{E}, \mathcal{L}_f, \mathcal{L}_s)$, where the labelling function $\mathcal{L}_f : \mathcal{V} \mapsto \mathcal{F}$ assigns each vertex $v_i \in \mathcal{V}$ to a feature $f_i = \mathcal{L}_f(v_i)$ in some feature space $(\mathcal{F}, d_f)$. Similarly, we denote $\mathcal{L}_s : \mathcal{V} \mapsto \mathcal{S}$ as a structure function which links each vertex $v_i \in \mathcal{V}$ with its structure information $s_i = \mathcal{L}_s(v_i)$, e.g., edge information, in some structure space $(\mathcal{S}, d_s)$. By associating a weight to each vertex, we allow the graph $\mathcal{G}$ to be represented by a fully supported mixing measure $\mu = \sum_{i=1}^N w_i \delta_{(f_i, s_i)}$ over the product between feature space and structure space $\mathcal{F} \times \mathcal{S}$. Notably, $\mu$ is not necessarily a probability measure as the summation of its weights can be different from one. We have the vertex affinity matrix between two graphs as $\boldsymbol{A}^v \in \mathbb{R}^{M \times N}$, where $A_{i,j}^v = (d_f(f_i, f_j))i, j$. Structural similarity is measured by pairwise distances within each graph,

represented by $\mathbf{A}^e \in \mathbb{R}^{|\mathcal{E}_1||\mathcal{E}_2|}$, with $\mathbf{A}^e_{i,j,k,l} = |d_s(s_i, s_k) - d_s(s_j, s_l)|$, where $d_s(\cdot)$ models node distance, such as the shortest path. We then define the space of all structured graphs $(\mathcal{F} \times \mathcal{S}, d_f, \mu)$ over a metric feature space $(\mathcal{F}, d_f)$ as $\mathbb{S}(\mathcal{F})$, where $(\mathcal{S}, d_s)$ is a metric structure space and $\mu = \sum_{i=1}^N w_i \delta_{(f_i, s_i)}$ is a mixing measure over $\mathcal{F} \times \mathcal{S}$.

**Theorem 1** (**Metric properties**). *The distance $d_{SoGA}$ in Eq. (8) defines a metric in $\mathbb{S}(\mathcal{F})$.*

Theorem 1 is proved in Appendix A.1. Intuitively, we can compare two graphs, $\mathcal{G}_1$ and $\mathcal{G}_2$, described by their mixing measures $\mu_1 = \sum_{i=1}^M w_{1i} \delta_{(f_{1i}, s_{1i})}$ and $\mu_2 = \sum_{j=1}^N w_{2j} \delta_{(f_{2j}, s_{2j})}$, with uniform vertex weights (*i.e.*, $w_{1i} = 1, \forall i \in [M]$, $w_{2j} = 1, \forall j \in [N]$) and shortest path structure matrices. The SoGA distance is zero if and only if there exists a one-to-one mapping between the graph vertices that preserves both shortest paths and features and both graphs have the same number of vertices.

Moreover, we also derive new constant speed geodesic in the SoGA context via Theorem 2, which is proved in Appendix A.2. This geodesic property is useful in optimal transport (discrete measures) to define the dynamic formulation and can be used in representation learning by sampling new data on the geodesic connect two endpoints graphs.

**Theorem 2** (**Geodesic space**). *The space $\mathbb{S}(\mathcal{F})$ equipped with the $d_{SoGA}$ distance is geodesic.*

## 4 EXPERIMENTS

### 4.1 IMPLEMENTATION DETAILS

**Model architectures.** We use the `LLaMA` large language model (Touvron et al., 2023), the `CLIP-ViT-L-Patch14` visual encoder (Radford et al., 2021), and an MLP projection similar to `LLaVA 1.5` (Liu et al., 2024a). Stage 1 follows the standard `LLaVA-Med` (Li et al., 2024) setup, while *stage 2 incorporates our multi-graph alignment with autoregressive training*. For multi-graph alignment, a 2-layer graph convolutional network is applied to the output of the Projection and LLM Decoder (handling both image and text modalities). We train for 1 epoch in stage 1 and 3 epochs in stage 2 using the same dataset as `LLava-Med`. The model is optimized using `Adam` (Kingma, 2014) with CosineAnnealingLR scheduler and learning rates of $2e-3$ and $2e-5$ for stages 1 and 2, respectively.

**Pre-training data.** We follow the same dataset being used in `LLaVA-Med` (Li et al., 2024). For stage 1, there are 600K image-text pairs filtered from `PMC-15M`. They are all converted to instruction-following data with simple instructions for describing the image. For stage 2, there are 60K image-text pairs extracted from `PMC-15M` with 5 modalities: CXR (chest X-ray), CT (computed tomography), MRI (magnetic resonance imaging), histopathology, and gross (i.e., macroscopic) pathology. Then, the authors from `LLaVA-Med` (Li et al., 2024) use language-only `GPT-4` to generate multi-round questions and answers in a tone as if it could see the image to convert these pairs to instruction-following format.

**Running-time.** We train LOGRA-MED using 4 A100-GPUs per with 80GB for both stages and complete the training process for stage 1 in 7 hours and for stage 2 in 7.5 hours. With original `LLaVA-Med` (version 1.5) (Li et al., 2024), the training process for stage 1 finishes in 6.5 hours, and for stage 2 finishes in 7 hours. In total, we need extra one more hour to complete the whole pre-training process compared to the LLaVa-Med.

### 4.2 DATA-HUNGRY PROBLEM OF AUTOREGRESSIVE

We begin by presenting the data-intensive nature of autoregressive training. To assess this, we used `LLaVA-Med`, a state-of-the-art multimodal large language model in the biomedical domain. `LLaVA-Med` follows a two-stage in their training process: Stage 1 aligns image-text tokens with biomedical concepts, and Stage 2 fine-tunes the model for instruction-based tasks. We pre-trained `LLaVA-Med` on varying data amounts (10%, 40%, 70%) and used the checkpoints to fine-tune the visual question answering downstream task. As shown in Figure 1, performance on the VQA-RAD dataset sharply declines when using 10% pre-training data compared with fully-training on 100%. To the best of our knowledge, this is the first time that the auto-regressive mechanism's data dependency issue in medical-MLLM has been revealed. This evidence indicates that without using enough instruction-tuning data, the auto-regressive creates a weak connection between visual features and text embeddings in LLM.

To tackle this issue, our LoGra-Med enhances image-text alignment through the semantic constraints between input images, their instruction response, and the long-context extended contexts. We experiment with LoGra-Med under the same settings as LLaVA-Med, pretraining on varying data sizes and utilizing the checkpoints for instruction fine-tuning. Figure 1 demonstrates that LoGra-Med effectively learns the biomedical concept alignment task, producing strong pretraining checkpoints for downstream fine-tuning while using only 10% of the data. For instance, while LLaVa-Med is at 52.39 with 10%, LoGra-Med reaches 72.52%, which is approximately LLaVa-Med pre-trained on 100%. Across 10%, 40%, and 70% data usage, LoGra-Med shows stable performance with consistent improvement, positioning it as a competitive model compared to other medical-MLLM like BioMed-GPT (Zhang et al., 2023a) or Med-MoE (Jiang et al., 2024). This suggests that the multi-graph alignment strategy effectively mitigates the data-intensive demands of autoregressive mechanisms in mLLMs.

## 4.3 Multi-modal Pre-training Comparison

To validate the benefit of our multi-graph alignment, we compare it with other vision-language pre-trained methods designed to train for the CLiP model or in visual instruction tuning to bridge frozen vision-language models.

| Method | VQA-RAD | | | SLAKE | | | PathVQA | | | Overall |
|---|---|---|---|---|---|---|---|---|---|---|
| | Open | Closed | Avg. | Open | Closed | Avg. | Open | Closed | Avg. | |
| LLaVA-Med (100%) | 63.65 | 81.62 | 72.64 | 83.44 | 83.41 | 83.43 | 36.78 | 91.33 | 64.06 | 73.37 |
| LLaVA-Med (10%) | 43.38↓20.27 | 61.4↓20.22 | 52.39↓20.25 | 80.94↓2.5 | 80.29↓3.12 | 80.62↓2.81 | 24.26↓13.69 | 88.03↓3.18 | 56.15↓7.91 | 63.05↓10.32 |
| InfoNCE | 59.39 | 77.57 | 68.48 | 82.4 | 83.17 | 82.78 | 34.59 | 91.45 | 63.02 | 71.43 |
| PLOT | 16.86 | 26.47 | 21.67 | 37.81 | 56.25 | 47.03 | 11.79 | 81.36 | 46.58 | 38.42 |
| SigLIP | 56.99 | 77.94 | 67.47 | 80.86 | 80.53 | 80.69 | 18.08 | 50.85 | 34.465 | 60.88 |
| VLAP | 57.49 | 76.47 | 66.98 | 80.05 | 82.21 | 81.13 | 32.21 | 91.16 | 61.685 | 69.93 |
| **LoGra-Med** | **66.02** | **79.04** | **72.52** | **84.92** | **85.1** | **85.01** | **37.25** | **91.45** | **64.34** | **73.96** |

Table 1: Fine-tuning performance on MedVQA downstream datasets (pre-trained 10%). **Bold** indicates the best values among pre-training algorithms, excluding LLaVA-Med (pre-trained 100%).

**Datasets.** We evaluate pre-trained models on three prominent biomedical VQA datasets: VQA-RAD (Lau et al., 2018), SLAKE (Liu et al., 2021), and PathVQA (He et al., 2020). VQA-RAD includes 3,515 questions across 315 radiology images, while SLAKE contains 642 radiology images from various body parts and over 7k QA pairs. PathVQA, focused on pathology, features 5k images and 32.8k questions. All datasets include open-ended (e.g., what, why, where) and closed-ended (yes/no or two-option) question types. We provide more details in the Appendix.

**Baselines.** We compare four approaches, including InfoNCE-based methods (Khan & Fu, 2023; Liu et al., 2023), SigLIP (Zhai et al., 2023), PLOT (Chen et al., 2022), and VLAP (Park et al., 2024). Among this, SigLIP adapts the Sigmoid loss on image-text pairs to break the global view of the pairwise similarities for normalization, resulting in scaling in large batch size. PLOT defines optimal transport as a distance between visual image patches and text embedding. In contrast, VLAP uses assignment prediction to bridge the modality gap between the visual and LLM embeddings. We train the baselines under the same settings as LoGra-Med with varying pre-training data rates and compare their performance on downstream tasks.

**Results.** In Tables 1, 2 and 12, We compare the performance of LoGra-Med and the baselines when pre-trained with 10%, 40%, and 100% of instruction-tuning data. While most contrastive baselines improve LLaVA-Med at 10%, LoGra-Med consistently outperforms LLaVA-Med and other methods overall. In open-ended questions, which require external knowledge, LoGra-Med delivers the best results. Notably, it maintains stable improvements across all three VQA datasets, unlike other methods that peak at 40% pre-training and decline afterward. For example, SigLIP scores 72.14% (Average) and 72.12% (Overall) on VQA-RAD at 40% but drops over 6% and 1% at 100%. In contrast, LoGra-Med continues to improve, reaching 74.91% (Average) and 74.75% (Overall).

| Method | VQA-RAD | | | SLAKE | | | PathVQA | | | Overall |
|---|---|---|---|---|---|---|---|---|---|---|
| | Open | Closed | Avg. | Open | Closed | Avg. | Open | Closed | Avg. | |
| LLaVA-Med (100%) | 63.65 | 81.62 | 72.64 | 83.44 | 83.41 | 83.43 | 36.78 | 91.33 | 64.06 | 73.37 |
| LLaVA-Med (40%) | 62.23↓1.42 | 79.41↓2.21 | 70.82↓1.82 | 84.42↑0.98 | 83.65↓0.24 | 84.04↑0.61 | 31.86↓4.92 | 84.99↓6.34 | 58.43↓5.63 | 71.09↓2.28 |
| InfoNCE | 63.11 | 77.57 | 70.34 | 82.68 | 83.89 | 83.29 | 33.58 | 89.62 | 61.6 | 71.74 |
| PLOT | 64.36 | 79.41 | 71.89 | 83.38 | 82.93 | 83.16 | 35.11 | 89.59 | 62.35 | 72.46 |
| SigLIP | 63.02 | 81.25 | 72.14 | 81.26 | 80.29 | 80.77 | 36.01 | 90.86 | 63.435 | 72.12 |
| VLAP | 63.17 | 79.04 | 71.11 | 83.38 | 83.89 | 83.64 | 35.62 | 90.83 | 63.225 | 72.66 |
| **LoGra-Med** | **66.01** | **82.72** | **74.37** | **84.17** | **85.82** | **84.99** | **37.41** | **91.27** | **64.34** | **74.57** |

Table 2: Performance fine-tuning on MedVQA downstream datasets (pre-trained 40%). **Bold** indicate for best values among pre-training algorithms excluding LLaVA-Med (pre-trained 100%).

## 4.4 MED-VQA COMPARISON WITH MEDICAL MLLMs

We now compare LOGRA-MED pre-trained with 100% data against other medical foundation models, each trained on varying datasets and employing different architectures or model sizes.

**Baselines.** We compare with *eight competitors*, both generic or medical foundation models, including LLaVA (Liu et al., 2024b), LLaVA-Med (Li et al., 2024), Med-Flamingo (Moor et al., 2023), Med-Dr (He et al., 2024), Biomed-GPT (Zhang et al., 2023a), M2I2 (Li et al., 2023b), GPT-4o (Achiam et al., 2023) and Med-MoE (Jiang et al., 2024). Whilst LLaVA and GPT-4o have no medical background, the others are pre-trained on a variety of biomedical knowledge. With the exception of LLaVa, which we reproduced, the results for the other baselines are taken from the literature. Moreover, we also present an enhanced version, LOGRA-MED + DCI, which integrates multi-scale visual features from vision encoders (Yao et al., 2024), potentially benefiting medical image analysis by considering both local (detailed) and global (contextual) features.

| Method | #Params | VQA-RAD | | | SLAKE | | | PathVQA | | | Overall |
|---|---|---|---|---|---|---|---|---|---|---|---|
| | | Open | Closed | Avg. | Open | Closed | Avg. | Open | Closed | Avg. | |
| LLaVA-Med (Li et al., 2024) | 7B | 63.65 | 81.62 | 72.64 | 83.44 | 83.41 | 83.43 | 36.78 | 91.33 | 64.06 | 73.37 |
| BiomedGPT-B (Zhang et al., 2023a) | 182M | 60.9 | 81.3 | 71.1 | 84.3 | 89.9 | 87.1 | 28 | 88 | 58 | 72.07 |
| M2I2 (Li et al., 2023b) | - | 61.8 | 81.6 | 71.7 | 74.7 | 91.1 | 82.9 | 36.3 | 88 | 62.15 | 72.25 |
| BioMed-CLIP (Zhang et al., 2023d) | 422M | 67.6 | 79.8 | 73.7 | 82.5 | 89.7 | 86.1 | | | | |
| Med-Dr (He et al., 2024) | 40B | 37.5 | 78.9 | 58.2 | 74.2 | 83.4 | 78.8 | 33.5 | 90.2 | 61.85 | 66.28 |
| LLaVA (general) (Liu et al., 2024b) | 7B | 50 | 65.1 | 57.55 | 78.2 | 63.2 | 70.7 | 7.7 | 63.2 | 35.45 | 54.57 |
| GPT-4 (Achiam et al., 2023) | 200B | 39.5 | 78.9 | 59.2 | 33.6 | 43.6 | 38.6 | | | | |
| Med-MoE (Phi2) (Jiang et al., 2024) | 3.6B | 58.55 | 82.72 | 70.64 | 85.06 | 85.58 | 85.32 | 34.74 | 91.98 | 63.36 | 73.11 |
| Med-MoE (Stable LM) (Jiang et al., 2024) | 2B | 50.08 | 80.07 | 65.08 | 83.16 | 83.41 | 83.29 | 33.79 | 91.30 | 62.55 | 70.3 |
| **LoGra-Med** | 7B | 66.35 | 83.46 | 74.91 | 85.34 | 85.58 | 85.46 | 36.82 | 90.92 | 63.87 | 74.75 |
| **LoGra-Med + DCI** | 7B | 67.31 | 81.99 | 74.65 | 84.88 | 85.82 | 85.23 | 37.77 | 91.86 | 64.82 | 74.9 |

Table 3: Performance comparing on MedVQA vs. other Med-MLLM.

**Results.** Overall, two LOGRA-MED versions perform better than the baseline models (Table 3), especially the version combined with DCI gains the best performance for PathVQA on Average score (64.82%) and for Overall result (74.9%). Compared to LLaVA-Med, LOGRA-MED exhibits significant improvements on every task result, such as 2.01% on VQA-RAD, 2.03% on SLAKE and 0.76% on PathVQA. Furthermore, it is important to note that both the plain and DCI versions of LOGRA-MED demonstrate competitive performance across all VQA datasets, despite having significantly fewer parameters. For instance, the two 7B-parameter LOGRA-MED versions outperform the 40B-parameter Med-Dr across all three datasets.

## 4.5 MEDICAL VISUAL CHATBOT EVALUATION

**Datasets.** Following the experimental settings in LLaVA-Med, we evaluate LOGRA-MED on a biomedical multimodal conversational dataset containing a total of 193 questions including conversation (143 samples), description (50 samples). Additionally, the experimental setup categorizes the test samples into five medical domains, including Chest X-ray (CXR), Magnetic Resonance Imaging (MRI), Histology, Gross, and Computed Tomography (CT).

**Baselines.** We evaluate with several SOTA multimodal large language models, including general models like LLaVA and GPT-4o, as well as medical-focused models such as LLaVA-Med and its variants, Med-Flamingo, Med-Dr, and Biomed-GPT. We use the officially provided weights for all comparison baselines without additional reproduction steps. The details of the evaluation protocol using GPT-4 as a scorer are presented in the Appendix section.

**Results.** Table 4 summarizes our finding results, which shows that in most of the settings, two of our LOGRA-MED deliver the top records. We provide additional analytics and visualizations of typical model outputs in the Appendix.

| Method | #Params | Question Type | | Domain | | | | | Overall |
|---|---|---|---|---|---|---|---|---|---|
| | | Conver. | Descr. | CXR | MRI | Histology | Gross | CT | |
| LLAVA | 7B | 39.40 | 26.20 | 41.60 | 33.40 | 38.40 | 32.91 | 33.40 | 36.1 |
| LLAVA-Med 1.0* | 7B | 47.4 | 33.99 | 51.31 | 36.32 | 45.61 | 41.09 | 44.87 | 43.93 |
| LLAVA-Med 1.5* | 7B | 46.78 | 34.58 | 54.58 | 36.5 | 41.85 | 40.3 | 45.02 | 43.62 |
| MedFlamingo | 8.3B | 28.58 | 13.89 | 26.93 | 21.34 | 22.09 | 32.71 | 22.25 | 24.77 |
| Med-Dr | 40B | 35.61 | 19.28 | 38.98 | 26.28 | 29.10 | 35.40 | 28.30 | 31.38 |
| Biomed-GPT | 182M | 20.71 | 17.99 | 27.53 | 18.50 | 17.18 | 14.72 | 22.08 | 20.01 |
| GPT-4o | 200B | 42.04 | 25.47 | 42.77 | 39.74 | 38.68 | 31.40 | 35.59 | 37.75 |
| **LoGra-Med** | 7B | 48.49 | 34.32 | 58.37 | 36.82 | 46.05 | 45.19 | 38.24 | 44.82 |
| **LoGra-Med + DCI** | 7B | 48.99 | 34.01 | 59.9 | 32.34 | 51.88 | 42.53 | 38.28 | 45.11 |

Table 4: Medical visual chatbot evaluation. Results are reported using GPT-4 as the scorer.

| Method | VQA-RAD | SLAKE |
|---|---|---|
| LoGra-Med (Full) | **74.37** | **84.99** |
| LVM-Med with synonyms | 72.39 | 82.93 |
| LoGra-Med in two stages | 72.81 | 84.14 |
| LVM-Med w/o long-context | 72.12 | 81.95 |
| LVM-Med w/o message passing | 73.90 | 84.29 |

Table 5: LoGra-Med ablation study. Results are presented as average scores on VQA-RAD and SLAKE, using pre-trained weights on 40%. The two key factors are emphasized.

### 4.6 ZERO-SHOT IMAGE CLASSIFICATION AS MEDVQA

**Datasets.** We assess the generalization of LOGRA-MED on zero-shot image classification by adapting public datasets from (Hu et al., 2024). We convert image classification tasks into visual question answering, where the model selects the correct answer from input options based on the given image (See Figure 6, Appendix). We focus on three key data modalities prevalent in our pre-training: `Microscopy`, Computed Tomography (`CT`), and Chest X-Ray (`CXR`). This evaluation spans several downstream tasks, including 8 datasets for `Microscopy`, 4 for `CT`, and 11 for `CXR`, totaling 23 datasets.

**Baselines.** We use checkpoints from `LLaVa-Med`, `Med-Flamingo`, and `RadFM` (Wu et al., 2023a) for zero-shot inference on the collected datasets. Notably, `RadFM` is pre-trained on $16M$ 2D and 3D medical scans, while LOGRA-MED is trained on just $600K$ instruction-following data. For baseline models, we follow the prompts proposed by (Hu et al., 2024), with detailed evaluations using third-party software to align model outputs with ground-truth answers, as outlined in the Appendix.

**Results.** Figure 4 illustrates the average performance of LOGRA-MED across Microscopy, CT, and Chest X-Ray modalities, with the total number of images and question-answer items listed below. Detailed results for each dataset within these modality groups are provided in Tables 8, 10, and 9 in the Appendix. Overall, LOGRA-MED still outperforms other models across all datasets, especially excelling in the microscopy modality, where it exceeds the runner-up, RadFM, by $8.2\%$. We attribute these benefits to the strong alignment between visual features and language embeddings achieved through triplet constraints, which compel the model to capture deeper semantic relationships.

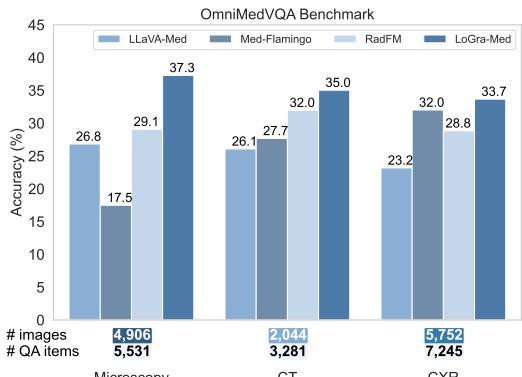

Figure 4: LOGRA-MED performance on 23 *zero-shot image classification tasks* within three data modalities.

### 4.7 ABLATION STUDIES

We validate LOGRA-MED under the following settings: (i) applying multi-graph alignment in both stages instead of only in stage 2 (default); (ii) performing simple paraphrasing during instruction tuning by replacing keywords with their synonyms; (iii) using only two graphs in the alignment step (excluding long-context graphs); and (iv) applying message passing to enhance node features. Table 5 shows that each proposed factor boosts the final performance wherein the multi-graph alignment and using ChatGPT as an extender are the most crucial components.

## 5 CONCLUSION

We have shown that enforcing triplet correlations among image modalities, their instruction data, and the extended contextual captions can improve vision-language alignment, which is often lacking in models trained by auto-regressive, especially given less pre-training data size as demonstrated for the LLaVa-Med model. We also present LOGRA-MED, a new multi-graph alignment algorithm to handle such requirements, which is efficient in training and matches LLAVA-Med's performance on just 10% of the training data while outperforming other state-of-the-art methods on various tasks. These findings underscore that selecting the appropriate learning algorithm for training MLLMs is as crucial as scaling model size or data volume.

**Limitations and Future Work** While we have conducted several experiments using the LLaVa model, it is crucial to validate LOGRA-MED with other architectures, such as the Flamingo model (Alayrac et al., 2022). Additionally, integrating a vision encoder or LLM specifically trained on medical sources (Chen et al., 2023b; MH Nguyen et al., 2024) holds promise for enhancing performance and improving robustness—an approach we have not yet evaluated. Lastly, we propose to investigate adaptor-based methods (Hu et al., 2022; Zhang et al., 2023b) for fine-tuning downstream tasks, as frozen backbone models can significantly reduce memory usage and maximize efficiency.

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

# SUPPLEMENTARY MATERIAL FOR "LoGra-Med: Long-Context Multi-Graph Learning for Medical Vision-Language Models"

## CONTENTS

## A  PROOFS OF THE MAIN THEORETICAL RESULTS

In this appendix, we provide detailed technical proofs of our main theoretical results.

### A.1  PROOF OF THEOREM 1

For the sake of simplicity, we denote the labeled graphs $\mathcal{G}$ and structured graphs discussed above only by $\mu$ the whole structured graph.

To prove Theorem 1, for any two graphs $\mathcal{G}_1$ and $\mathcal{G}_2$ in the structured graph space $\mathbb{S}(\mathcal{F})$, described respectively by their mixing measure $\mu_1 = \sum_{i=1}^{M} w_{1i}\delta_{(f_{1i},s_{1i})}$ and $\mu_2 = \sum_{j=1}^{N} w_{2j}\delta_{(f_{2j},s_{2j})}$, respectively, we wish to prove the following properties:

1. Positivity: $d_{\text{SoGA}}(\mathcal{G}_1, \mathcal{G}_2) > 0$ for any $\mathcal{G}_1 \neq \mathcal{G}_2$.
2. Equality relation: $d_{\text{SoGA}}(\mathcal{G}_1, \mathcal{G}_2) = 0$ if and only if $\mathcal{G}_1 = \mathcal{G}_2$.
3. Symmetry: $d_{\text{SoGA}}(\mathcal{G}_1, \mathcal{G}_2) = d_{\text{SoGA}}(\mathcal{G}_2, \mathcal{G}_1)$.
4. Triangle inequality: $d_{\text{SoGA}}(\mathcal{G}_1, \mathcal{G}_3) \leq d_{\text{SoGA}}(\mathcal{G}_1, \mathcal{G}_2) + d_{\text{SoGA}}(\mathcal{G}_2, \mathcal{G}_3)$ for any graph $\mathcal{G}_3$.

Note first that 1. Positivity and 3. Symmetry hold trivially.

**Proof of 2. Equality relation.** The equality relation immediately follows the following Proposition 1, which is proved in Appendix B.1.

**Proposition 1** (Equality relation). *For any two graphs $\mathcal{G}_1$ and $\mathcal{G}_2$ in the structured graph space $\mathbb{S}(\mathcal{F})$, described respectively by their mixing measure $\mu_1 = \sum_{i=1}^{M} w_{1i}\delta_{(f_{1i},s_{1i})}$ and $\mu_2 = \sum_{j=1}^{N} w_{2j}\delta_{(f_{2j},s_{2j})}$, it holds $d_{SoGA}(\mathcal{G}_1, \mathcal{G}_2) = 0$ if and only if $M = N$ and there exists a bijection $\sigma : [M] \mapsto [M]$ such that:*

*E1.* $\forall i \in [M] : w_{1i} = w_{2\sigma(i)}$.

*E2.* $\forall i \in [M] : f_{1i} = f_{2\sigma(i)}$.

*E3.* $\forall i, k \in [M]^2 : d_s(s_{1i}, s_{1k}) = d_s(s_{2\sigma(i)}, s_{2\sigma(k)})$.

**Proof of 4. Triangle inequality.** Let us consider two arbitrary graphs $\mathcal{G}_1$ and $\mathcal{G}_2$, described respectively by their probability measure $\mu_1 = \sum_{i=1}^{M} w_{1i}\delta_{(f_{1i}, s_{1i})}$ and $\mu_2 = \sum_{j=1}^{N} w_{2j}\delta_{(f_{2j}, s_{2j})}$. For any graph $\mathcal{G}_3$ described by its probability measure $\mu_3 = \sum_{i=1}^{K} w_{3k}\delta_{(f_{3k}, s_{3k})}$, we define $\boldsymbol{P} \in \mathcal{A}(\mathcal{G}_1, \mathcal{G}_2)$ and $\boldsymbol{Q} \in \mathcal{A}(\mathcal{G}_2, \mathcal{G}_3)$ as two optimal couplings of the SoGA distance between $\mu_1$ and $\mu_2$ and $\mu_2$ and $\mu_3$, respectively, *i.e.*,

$$\boldsymbol{P} \in \mathcal{A}(\mathcal{G}_1, \mathcal{G}_2) = \left\{ \boldsymbol{P} \in \{0,1\}^{M \times N} : \sum_{i=1}^{M} P_{i,j} = w_{1j} = 1, \sum_{j=1}^{N} P_{i,j} = w_{2i} = 1 \right\},$$

$$\boldsymbol{Q} \in \mathcal{A}(\mathcal{G}_2, \mathcal{G}_3) = \left\{ \boldsymbol{Q} \in \{0,1\}^{N \times K} : \sum_{j=1}^{N} Q_{j,k} = w_{2k} = 1, \sum_{k=1}^{K} Q_{j,k} = w_{3j} = 1 \right\}.$$

We then construct $\boldsymbol{R} = \left( \sum_j \frac{P_{i,j}Q_{j,k}}{w_{2j}} \right)_{i,k}$. Then it holds that $\boldsymbol{R} \in \mathcal{A}(\mathcal{G}_1, \mathcal{G}_3)$. Indeed, we have

$$\sum_i R_{i,k} = \sum_i \sum_j \frac{P_{i,j}Q_{j,k}}{w_{2j}} = \sum_j \sum_i P_{i,j} \frac{Q_{j,k}}{w_{2j}} = \sum_j w_{1j} \frac{Q_{j,k}}{w_{2j}} = \sum_j Q_{j,k} = 1.$$

By the suboptimality of $\boldsymbol{R}$, the triangle inequalities of $d_f$ and $|\cdot|$, we have

$$d_{\text{SoGA}}(\mathcal{G}_1, \mathcal{G}_3) \leq \sum_{i,j,k,l} \left[ d_f(f_{1i}, f_{3j}) + |d_s(s_{1i}, s_{1k}) - d_s(s_{3j}, s_{3l})| \right] R_{i,j}R_{k,l}$$

$$= \sum_{i,j,k,l} \left[ d_f(f_{1i}, f_{3j}) + |d_s(s_{1i}, s_{1k}) - d_s(s_{3j}, s_{3l})| \right] \sum_t \frac{P_{i,t}Q_{t,j}}{w_{2t}} \sum_d \frac{P_{k,d}Q_{d,l}}{w_{2d}}$$

$$= \sum_{i,j,k,l,t,d} \left[ d_f(f_{1i}, f_{3j}) + |d_s(s_{1i}, s_{1k}) - d_s(s_{3j}, s_{3l})| \right] \frac{P_{i,t}Q_{t,j}}{w_{2t}} \frac{P_{k,d}Q_{d,l}}{w_{2d}}$$

$$\leq \sum_{i,j,k,l,t,d} \left[ d_f(f_{1i}, f_{2t}) + d_f(f_{2t}, f_{3j}) \right] \frac{P_{i,t}Q_{t,j}}{w_{2t}} \frac{P_{k,d}Q_{d,l}}{w_{2d}}$$

$$+ \sum_{i,j,k,l,t,d} \left[ |d_s(s_{1i}, s_{1k}) - d_s(s_{2t}, s_{2d})| + |d_s(s_{2t}, s_{2d}) - d_s(s_{3j}, s_{3l})| \right] \frac{P_{i,t}Q_{t,j}}{w_{2t}} \frac{P_{k,d}Q_{d,l}}{w_{2d}}$$

$$= \sum_{i,j,k,l,t,d} \left[ d_f(f_{1i}, f_{2t}) + |d_s(s_{1i}, s_{1k}) - d_s(s_{2t}, s_{2d})| \right] \frac{P_{i,t}P_{k,d}}{w_{2t}} \frac{Q_{t,j}Q_{d,l}}{w_{2d}}$$

$$+ \sum_{i,j,k,l,t,d} \left[ d_f(f_{2t}, f_{3j}) + |d_s(s_{2t}, s_{2d}) - d_s(s_{3j}, s_{3l})| \right] \frac{P_{i,t}Q_{t,j}}{w_{2t}} \frac{P_{k,d}Q_{d,l}}{w_{2d}}$$

$$= \sum_{i,k,t,d} \left[ d_f(f_{1i}, f_{2t}) + |d_s(s_{1i}, s_{1k}) - d_s(s_{2t}, s_{2d})| \right] P_{i,t}P_{k,d} \sum_j \frac{Q_{t,j}}{w_{2t}} \sum_l \frac{Q_{d,l}}{w_{2d}}$$

$$+ \sum_{j,l,t,d} \left[ d_f(f_{2t}, f_{3j}) + |d_s(s_{2t}, s_{2d}) - d_s(s_{3j}, s_{3l})| \right] Q_{t,j}Q_{d,l} \sum_i \frac{P_{i,t}}{w_{2t}} \sum_k \frac{P_{k,d}}{w_{2d}}.$$

Note that we have

$$\sum_j \frac{Q_{t,j}}{w_{2t}} = \sum_l \frac{Q_{d,l}}{w_{2d}} = \sum_i \frac{P_{i,t}}{w_{2t}} = \sum_k \frac{P_{k,d}}{w_{2d}} = 1.$$

This is how we achieve the desired result, because

$$d_{\text{SoGA}}(\mathcal{G}_1, \mathcal{G}_3) \leq \sum_{i,k,t,d} \left[ d_f(f_{1i}, f_{2t}) + |d_s(s_{1i}, s_{1k}) - d_s(s_{2t}, s_{2d})| \right] P_{i,t} P_{k,d}$$

$$+ \sum_{j,l,t,d} \left[ d_f(f_{2t}, f_{3j}) + |d_s(s_{2t}, s_{2d}) - d_s(s_{3j}, s_{3l})| \right] Q_{t,j} Q_{d,l}$$

$$= d_{\text{SoGA}}(\mathcal{G}_1, \mathcal{G}_2) + d_{\text{SoGA}}(\mathcal{G}_2, \mathcal{G}_3) \text{ (since } \boldsymbol{P} \text{ and } \boldsymbol{Q} \text{ are the optimal plans)}.$$

## A.2 PROOF OF THEOREM 2

Theorem 2 enables us to characterise the optimal transport problem between two measures as a curve in the space of measures, with the objective of minimising its total length. Furthermore, this formulation is beneficial for deriving global minima results for non-convex particles in gradient descent in an optimisation context, which is a valuable application of gradient flows (Chizat & Bach, 2018). By definition, a geodesic between $\mathcal{G}_1$ and $\mathcal{G}_2$ is a shortest path between these two graphs. In particular, the computation of distances along constant speed geodesic paths is a relatively straightforward process, as these paths are directly embedded into the real line $\mathbb{R}$ as follows: $d_{\text{SoGA}}(\mathcal{G}_1, \mathcal{G}_2) = |t - u|^{-1} d_{\text{SoGA}}(p(u), p(t))$, for all $0 \leq u \neq t \leq 1$ and for any path (continuous map) $p$ connect $\mathcal{G}_1$ to $\mathcal{G}_2$ such that $p(u) = \mathcal{G}_1$ and $p(t) = \mathcal{G}_2$. To prove Theorem A.2, it is necessary to collect fundamental material using Definition 1 from metric geometry for a general metric space $(\mathbb{M}, d)$.

**Definition 1** (Length and geodesic spaces). *Let $(\mathbb{M}, d)$ be a metric space and two points $x, y \in \mathbb{M}$. We say that a path (curve) $p : [0, 1] \mapsto \mathbb{M}$ connect or join $x$ to $y$ if $p(0) = x$ and $p(1) = y$ and $p$ is a continuous map.*

*We also define the length $L(p) \in \mathbb{R}$ of a path $p : [0, 1] \mapsto \mathbb{M}$ as*

$$L(p) := \sup \sum_{i=1}^n d(p(t_i), p(t_{i+1}))$$

*where we take the supremum over all $n \geq 1$ and all $n$-tuples $t_1 < \ldots < t_n$ in $[0, 1]$.*

*We denote a metric space $\mathbb{M}$ as a length space if for all $x, y \in \mathbb{M}$, $d(x, y) = \inf_p L(p)$ where the infimum is taken over all paths $p$ connecting $x$ to $y$.*

*We call a length space as a geodesic space if for all $x, y \in \mathbb{M}$, there exists a path $p(x, y) : [0, 1] \mapsto \mathbb{M}$ such that*

$$d(x, y) = \min_{p(x,y)} L(p(x, y)).$$

*We also denote the path $p(x, y)$ as a geodesic between $x$ and $y$.*

*Finally, we define a path $p : [0, 1] \mapsto \mathbb{M}$ as a constant speed geodesic if and only if*

$$d(p(u), p(t)) = |t - u| d(p(0), p(1)), \forall u, t \in [0, 1].$$

For the proof of Theorem 2, we first consider an optimal coupling $\boldsymbol{V}^*$ for SoGA distance between two graphs $\mathcal{G}_1$ and $\mathcal{G}_2$, *i.e.,*

$$d_{\text{SoGA}}(\mathcal{G}_1, \mathcal{G}_2) = \min_{\boldsymbol{V} \in \mathcal{A}(\mathcal{G}_1, \mathcal{G}_2)} O(\boldsymbol{A}^v, \mathbf{A}^e, \boldsymbol{V}) = O(\boldsymbol{A}^v, \mathbf{A}^e, \boldsymbol{V}^*),$$

described respectively by their mixing measure $\mu_0 = \sum_{i=1}^M w_{0i} \delta_{(f_{0i}, s_{0i})}$ and $\mu_1 = \sum_{j=1}^N w_{1j} \delta_{(f_{1j}, s_{1j})}$. Moreover, for any $t \in [0, 1]$ we define $\nu_t : \mathcal{F} \times \mathcal{S}_0 \times \mathcal{F} \times \mathcal{S}_1 \mapsto \mathcal{F} \times \mathcal{S}_0 \times \mathcal{S}_1$ such that

$$\nu_t(f_0, s_0, f_1, s_1) = ((1-t)f_0 + t f_1, s_0, s_1), \text{ and } \mu_t := \nu_t \# \boldsymbol{V}^* = \sum_{i=1}^M \sum_{j=1}^N V_{i,j}^* \delta_{((1-t)f_0 + t f_1, s_{0i}, s_{1j})},$$

and on the metric space $\mathcal{S}_0 \times \mathcal{S}_1$, we define the distance

$$d_t := (1-t)d_{s_0} \oplus t d_{s_1} : (1-t)d_{s_0} \oplus t d_{s_1}((s_{0i}, s_{0j}), (s_{1k}, s_{1l})) = (1-t)d_s(s_{0i}, s_{1k}) + t d_s(s_{0j}, s_{1l})$$

for any $((s_{0i}, s_{0j}), (s_{1k}, s_{1l})) \in \mathcal{S}_0 \times \mathcal{S}_1$. Here, we denote $\#$ the push-forward operator such that $\nu_t \# \boldsymbol{V}^*(\mathbb{A}) = \boldsymbol{V}^*(\nu_t^{-1}(\mathbb{A}))$ for any Borel sets of a $\sigma$-albegra. For simplicity, we only consider $(\mathcal{F}, d_f) = (\mathbb{R}^d, \|\cdot\|)$ where $\|\cdot\|$ is the Euclidean norm.

Then we aim to prove that $(\mathcal{F} \times \mathcal{S}_0 \times \mathcal{S}_1, (1-t)d_{s_0} \oplus td_{s_1}, \mu_t)_{t \in [0,1]}$ is a constant speed geodesic joining $(\mathcal{F} \times \mathcal{S}_0, d_{s0}, \mu_0))$ and $(\mathcal{F} \times \mathcal{S}_1, d_{s1}, \mu_1))$, for arbitrary elements $(\mathcal{F} \times \mathcal{S}_0, d_{s0}, \mu_0))$ and $(\mathcal{F} \times \mathcal{S}_1, d_{s1}, \mu_1))$ in the metric space $(\mathbb{S}(\mathcal{F}), d_{\text{SoGA}})$.

To do so, we consider any $u, t \in [0,1]$ such that $u \neq t$. By definition, we have to prove that

$$d_{\text{SoGA}}(\mu_u, \mu_t) = |t - u| d_{\text{SoGA}}(\mu_0, \mu_1). \tag{9}$$

Indeed, to prove equation (9), we first recall that

$$\mu_u := \nu_u \# \boldsymbol{V}^* = \sum_{i=1}^{M} \sum_{j=1}^{N} V_{i,j}^* \delta_{((1-u)f_0 + uf_1, s_{0i}, s_{1j})},$$

$$\mu_t := \nu_t \# \boldsymbol{V}^* = \sum_{i=1}^{M} \sum_{j=1}^{N} V_{i,j}^* \delta_{((1-t)f_0 + tf_1, s_{0i}, s_{1j})},$$

$$d_{\text{SoGA}}(\mu_0, \mu_1) = \sum_{i,j,k,l} [d_f(f_{0i}, f_{1j}) + |d_s(s_{0i}, s_{1k}) - d_s(s_{0j}, s_{1l})|] V_{i,j}^* V_{k,l}^*.$$

We then define the coupling $\gamma^{u,t} = (\mu_u \times \mu_t) \# \boldsymbol{V}^* \in \mathcal{A}(\mu_u, \mu_t)$. By the suboptimality of $\gamma^{u,t}$, it holds that:

$$d_{\text{SoGA}}(\mu_u, \mu_t) \leq \sum_{i,j,k,l} [d_f(f_{0i}, f_{1j}) + |d_t((s_{0i}, s_{0j}), (s_{1k}, s_{1l})) - d_u((s_{0i}, s_{0j}), (s_{1k}, s_{1l}))|] \gamma_{i,j}^{u,t} \gamma_{k,l}^{u,t}$$

$$= \sum_{i,j,k,l} \Big[ d_f((1-t)f_{0i} + tf_{1j}, (1-u)f_{0i} + uf_{1j})$$

$$+ |(1-t)d_s(s_{0i}, s_{1k}) + td_s(s_{0j}, s_{1l}) - (1-u)d_s(s_{0i}, s_{1k}) - ud_s(s_{0j}, s_{1l})| \Big] V_{i,j}^* V_{k,l}^*$$

$$= \sum_{i,j,k,l} [(t-u)d_f(f_{0i}, f_{1j}) + |(t-u)d_s(s_{0i}, s_{1k}) - (t-u)d_s(s_{0j}, s_{1l})|] V_{i,j}^* V_{k,l}^*$$

$$= |t-u| \sum_{i,j,k,l} [d_f(f_{0i}, f_{1j}) + |d_s(s_{0i}, s_{1k}) - d_s(s_{0j}, s_{1l})|] V_{i,j}^* V_{k,l}^*$$

$$= |t-u| d_{\text{SoGA}}(\mu_0, \mu_1).$$

Here, we used the fact that $d_f$ is the Euclidean norm, hence
$$d_f((1-t)f_{0i} + tf_{1j}, (1-u)f_{0i} + uf_{1j}) = \|(1-t)f_{0i} + tf_{1j} - (1-u)f_{0i} - uf_{1j}\| = |t-u| d_f(f_{0i}, f_{1j}).$$
Therefore, we have

$$d_{\text{SoGA}}(\mu_u, \mu_t) \leq |t-u| d_{\text{SoGA}}(\mu_0, \mu_1). \tag{10}$$

The remaining task is to prove that

$$d_{\text{SoGA}}(\mu_u, \mu_t) \geq |t-u| d_{\text{SoGA}}(\mu_0, \mu_1). \tag{11}$$

To show that this inequality, we note that via the triangle inequality of $d_{\text{SoGA}}$ and for any $0 \leq u \leq t \leq 1$, it holds that

$$d_{\text{SoGA}}(\mu_0, \mu_1) \leq d_{\text{SoGA}}(\mu_0, \mu_u) + d_{\text{SoGA}}(\mu_u, \mu_t) + d_{\text{SoGA}}(\mu_t, \mu_1)$$

$$\leq u d_{\text{SoGA}}(\mu_0, \mu_1) + (t-u) d_{\text{SoGA}}(\mu_0, \mu_1) + (1-t) d_{\text{SoGA}}(\mu_0, \mu_1)$$

$$= d_{\text{SoGA}}(\mu_0, \mu_1).$$

Hence, for any $0 \leq u \leq t \leq 1$, we obtain

$$d_{\text{SoGA}}(\mu_0, \mu_u) + d_{\text{SoGA}}(\mu_u, \mu_t) + d_{\text{SoGA}}(\mu_t, \mu_1)$$

$$= u d_{\text{SoGA}}(\mu_0, \mu_1) + (t-u) d_{\text{SoGA}}(\mu_0, \mu_1) + (1-t) d_{\text{SoGA}}(\mu_0, \mu_1). \tag{12}$$

Suppose that

$$d_{\text{SoGA}}(\mu_u, \mu_t) < (t-u) d_{\text{SoGA}}(\mu_0, \mu_1).$$

Then combining with the fact that

$$d_{\text{SoGA}}(\mu_0, \mu_u) \leq u d_{\text{SoGA}}(\mu_0, \mu_1), \text{ and } d_{\text{SoGA}}(\mu_t, \mu_1) \leq (1-t) d_{\text{SoGA}}(\mu_0, \mu_1),$$

we have

$$d_{\text{SoGA}}(\mu_0, \mu_u) + d_{\text{SoGA}}(\mu_u, \mu_t) + d_{\text{SoGA}}(\mu_t, \mu_1)$$

$$< u d_{\text{SoGA}}(\mu_0, \mu_1) + (t-u) d_{\text{SoGA}}(\mu_0, \mu_1) + (1-t) d_{\text{SoGA}}(\mu_0, \mu_1).$$

This leads to the contradiction with the equation (12.) Hence the desired inequality in (11) holds. Finally, we obtain

$$d_{\text{SoGA}}(\mu_u, \mu_t) = |t - u| d_{\text{SoGA}}(\mu_0, \mu_1). \tag{13}$$

# B  PROOFS OF TECHNICAL RESULTS

## B.1  PROOF OF PROPOSITION 1

First, let us suppose that $d_{\text{SoGA}}(\mathcal{G}_1, \mathcal{G}_2) = 0$. We wish to prove the existence of a bijection $\sigma$ satisfying E1, E2, and E3. Indeed, let $\boldsymbol{V} \in \mathcal{A}(\mathcal{G}_1, \mathcal{G}_2)$ be any admissible transportation plan that encode a valid matching between $\mathcal{G}_1$ and $\mathcal{G}_2$. Then we define:

$$d(s_{1i}, s_{1k}) = \frac{1}{2}\left[d_f(f_{1i}, f_{1k}) + d_s(s_{1i}, s_{1k})\right], \quad \forall i, k \in [M]^2, \tag{14}$$

$$d(s_{2j}, s_{2l}) = \frac{1}{2}\left[d_f(f_{2j}, f_{2l}) + d_s(s_{2j}, s_{2l})\right], \quad \forall j, l \in [M]^2. \tag{15}$$

Recall that we then define SGM discrepancy as:

$$d_{\text{SoGA}}(\mathcal{G}_1, \mathcal{G}_2) = \min_{\boldsymbol{V} \in \mathcal{A}(\mathcal{G}_1, \mathcal{G}_2)}\left(\sum_{i,j} A_{i,j}^v V_{i,j} + \sum_{i,j,k,l} \boldsymbol{A}_{i,j,k,l}^e V_{i,j} V_{k,l}\right) = \min_{\boldsymbol{V} \in \mathcal{A}(\mathcal{G}_1, \mathcal{G}_2)} O(\boldsymbol{A}^v, \mathbf{A}^e, \boldsymbol{V})$$

$$= \min_{\boldsymbol{V} \in \mathcal{A}(\mathcal{G}_1, \mathcal{G}_2)}\left[O_f(\boldsymbol{A}^v, \boldsymbol{V}) + O_s(\mathbf{A}^e, \boldsymbol{V})\right]. \tag{16}$$

It should be recalled that the vertex affinity matrix $\boldsymbol{A}^v \in \mathbb{R}^{M \times M}$, defined as $A_{i,j}^v = (d_f(f_{1i}, f_{2j}))_{i,j}$, was introduced in the previous section. The edge affinity tensor, denoted by $\mathbf{A}^e$, is defined as follows: $\boldsymbol{A}_{i,j,k,l}^e = |d_s(s_{1i}, s_{1k}) - d_s(s_{2j}, s_{2l})|$.

Let $\boldsymbol{V}^*$ be the optimal coupling for $d_{\text{SoGA}}(\mathcal{G}_1, \mathcal{G}_2)$. Then we have

$$O_f(\boldsymbol{A}^v, \boldsymbol{V}^*) + O_s(\mathbf{A}^e, \boldsymbol{V}^*) = \min_{\boldsymbol{V} \in \mathcal{A}(\mathcal{G}_1, \mathcal{G}_2)}\left[O_f(\boldsymbol{A}^v, \boldsymbol{V}) + O_s(\mathbf{A}^e, \boldsymbol{V})\right] = d_{\text{SoGA}}(\mathcal{G}_1, \mathcal{G}_2) = 0. \tag{17}$$

Since both $O_f(\boldsymbol{A}^v, \boldsymbol{V}^*)$ and $O_s(\mathbf{A}^e, \boldsymbol{V}^*)$ are non-negative, we conclude that $O_f(\boldsymbol{A}^v, \boldsymbol{V}^*) = O_s(\mathbf{A}^e, \boldsymbol{V}^*) = 0$. Now we wish to use the following Lemma B.2, which is proved in Appendix B.2.

**Lemma 1.** *Given the definition of* $\overline{A}_{i,j,k,l}^e = |d(s_{1i}, s_{1k}) - d(s_{2j}, s_{2l})|$ *where* $d(s_{1i}, s_{1k})$ *and* $d(s_{2j}, s_{2l})$ *are provided in equations (14) and (15), respectively, it holds that*

$$O_s(\overline{\mathbf{A}}^e, \boldsymbol{V}^*) = \sum_{i,j,k,l} \overline{A}_{i,j,k,l}^e V_{i,j}^* V_{k,l}^* = \sum_{i,j,k,l} |d(s_{1i}, s_{1k}) - d(s_{2j}, s_{2l})| V_{i,j}^* V_{k,l}^* = 0. \tag{18}$$

*Moreover, there exists a bijective* $\sigma : [M] \mapsto [N]$ *with* $M = N$ *satisfies the weight and distance* $d$ *preserving isometry as follows:*

*E1.* $\forall i \in [M] : w_{1i} = w_{2\sigma(i)}.$

*E3\*.* $\forall i, k \in [M]^2 : d(s_{1i}, s_{1k}) = d(s_{2\sigma(i)}, s_{2\sigma(k)}).$

Because we have $\boldsymbol{V}^*$ is the optimal coupling w.r.t. the distance $d$ such that

$$O_s(\overline{\mathbf{A}}^e, \boldsymbol{V}^*) = \min_{\boldsymbol{V} \in \mathcal{A}(\mathcal{G}_1, \mathcal{G}_2)} O_s(\overline{\mathbf{A}}^e, \boldsymbol{V}) = 0, \tag{19}$$

$\boldsymbol{V}^*$ is supported by $\sigma$ and satisfies $\boldsymbol{V}^* = \boldsymbol{I}_{M \times N} \times \sigma$. Therefore, $O_f(\boldsymbol{A}^v, \boldsymbol{V}^*) = \sum_{i,j} d_f(f_{1i}, f_{2\sigma(i)}) V_{i,j}^* = \sum_i d_f(f_{1i}, f_{2\sigma(i)}) \sum_j V_{i,j}^* = \sum_i d_f(f_{1i}, f_{2\sigma(i)}) = 0$. Here, we used the fact that

$$\boldsymbol{V}^* \in \mathcal{A}(\mathcal{G}_1, \mathcal{G}_2) = \left\{\boldsymbol{V} \in \{0,1\}^{M \times N} : \sum_{i=1}^M V_{i,j} = w_{1j} = 1, \sum_{j=1}^N V_{i,j} = w_{2i} = 1\right\}.$$

Note that $d_f(f_{1i}, f_{2\sigma(i)}), i \in [M]$ are all non-negative. This leads to $d_f(f_{1i}, f_{2\sigma(i)}) = 0, \forall i \in [M]$. This is equivalent to $f_{1i} = f_{2\sigma(i)}, \forall i \in [M]$ since $d_f$ is a metric, which is the desired E2. Therefore,

we also have $d_f(f_{1i}, f_{1k}) = d_f(f_{2\sigma(i)}, f_{2\sigma(k)}), \forall i, k \in [M]$. Combining equations (14), (15), and E3*, we have

$$d(s_{1i}, s_{1k}) = \frac{1}{2} \left[ d_f(f_{1i}, f_{1k}) + d_s(s_{1i}, s_{1k}) \right], \tag{20}$$

$$d(s_{2\sigma(i)}, s_{\sigma(k)}) = \frac{1}{2} \left[ d_f(f_{2\sigma(i)}, f_{2\sigma(k)}) + d_s(s_{2\sigma(i)}, s_{2\sigma(k)}) \right], \quad \forall i, k \in [M]^2. \tag{21}$$

This leads to the desired result, *i.e.*, E3. $d_s(s_{1i}, s_{1k}) = d_s(s_{2\sigma(i)}, s_{2\sigma(k)}), \forall i, k \in [M]^2$.

Now, let us suppose that $M = N$ there exists a bijection $\sigma : [M] \mapsto [N]$ satisfying E1, E2, and E3. We wish to prove that $d_{\mathrm{SoGA}}(\mathcal{G}_1, \mathcal{G}_2) = 0$. Then we can consider the transportation plan $\boldsymbol{V}^* = \boldsymbol{I}_{M \times N} \times \sigma$, *i.e.*, $\boldsymbol{V}^*$ is associated with $i \mapsto i$ and $j \mapsto \sigma(i)$. Using E1, it holds that $\boldsymbol{V}^* \in \mathcal{A}(\mathcal{G}_1, \mathcal{G}_2)$. Moreover, via E2 and E3, we also have

$$d_{\mathrm{SoGA}}(\mathcal{G}_1, \mathcal{G}_2) = \min_{\boldsymbol{V} \in \mathcal{A}(\mathcal{G}_1, \mathcal{G}_2)} O(\boldsymbol{A}^v, \mathbf{A}^e, \boldsymbol{V}) \leq \sum_{i,j} A_{i,j}^v V_{i,j}^* + \sum_{i,j,k,l} \boldsymbol{A}_{i,j,k,l}^e V_{i,j}^* V_{k,l}^*$$

$$= \sum_{i,j} d_f(f_{1i}, f_{2j}) V_{i,j}^* + \sum_{i,j,k,l} |d_s(s_{1i}, s_{1k}) - d_s(s_{2j}, s_{2l})| V_{i,j}^* V_{k,l}^*$$

$$= \sum_{i,j} d_f(f_{1i}, f_{2\sigma(i)}) V_{i,j}^* + \sum_{i,j,k,l} |d_s(s_{1i}, s_{1k}) - d_s(s_{2\sigma(i)}, s_{2\sigma(k)})| V_{i,j}^* V_{k,l}^* = 0.$$

This leads to the desired result that $d_{\mathrm{SoGA}}(\mathcal{G}_1, \mathcal{G}_2) = 0$.

### B.2 PROOF OF LEMMA 1

By definitions and the triangle inequalities of the metric $d_f$ and $d_s$, we have

$$O_s(\overline{\mathbf{A}}^e, \boldsymbol{V}^*) = \sum_{i,j,k,l} |d(s_{1i}, s_{1k}) - d(s_{2j}, s_{2l})| V_{i,j}^* V_{k,l}^*$$

$$= \sum_{i,j,k,l} |\frac{1}{2} \left[ d_f(f_{1i}, f_{1k}) + d_s(s_{1i}, s_{1k}) \right] - \frac{1}{2} \left[ d_f(f_{2j}, f_{2l}) + d_s(s_{2j}, s_{2l}) \right] | V_{i,j}^* V_{k,l}^*$$

$$= \sum_{i,j,k,l} \left| \frac{1}{2} \left[ d_f(f_{1i}, f_{1k}) - d_f(f_{2j}, f_{2l}) \right] + \frac{1}{2} \left[ d_s(s_{1i}, s_{1k}) - d_s(s_{2j}, s_{2l}) \right] \right| V_{i,j}^* V_{k,l}^*$$

$$\leq \frac{1}{2} \sum_{i,j,k,l} |d_f(f_{1i}, f_{1k}) - d_f(f_{2j}, f_{2l})| V_{i,j}^* V_{k,l}^* + \frac{1}{2} \sum_{i,j,k,l} |d_s(s_{1i}, s_{1k}) - d_s(s_{2j}, s_{2l})| V_{i,j}^* V_{k,l}^*$$

$$= \frac{1}{2} \sum_{i,j,k,l} |d_f(f_{1i}, f_{1k}) - d_f(f_{2j}, f_{2l})| V_{i,j}^* V_{k,l}^* + \frac{1}{2} O_s(\mathbf{A}^e, \boldsymbol{V}^*)$$

$$= \frac{1}{2} \sum_{i,j,k,l} |d_f(f_{1i}, f_{1k}) - d_f(f_{2j}, f_{2l})| V_{i,j}^* V_{k,l}^* \text{ (since } O_s(\mathbf{A}^e, \boldsymbol{V}^*) = 0). \tag{22}$$

Using the triangle inequality of the metric $d_f$ again, we have

$$d_f(f_{1i}, f_{1k}) \leq d_f(f_{1i}, f_{2j}) + d_f(f_{2j}, f_{2l}) + d_f(f_{2l}, f_{1k}),$$
$$d_f(f_{2j}, f_{2l}) \leq d_f(f_{2j}, f_{1i}) + d_f(f_{1i}, f_{1k}) + d_f(f_{1k}, f_{2l}).$$

This is equivalent to

$$d_f(f_{1i}, f_{1k}) - d_f(f_{2j}, f_{2l}) \leq d_f(f_{1i}, f_{2j}) + d_f(f_{1k}, f_{2l}),$$
$$d_f(f_{2j}, f_{2l}) - d_f(f_{1i}, f_{1k}) \leq d_f(f_{1i}, f_{2j}) + d_f(f_{1k}, f_{2l}). \tag{23}$$

We consider two sets $I_1 = \{i, j, k, l : d_f(f_{1i}, f_{1k}) - d_f(f_{2j}, f_{2l}) \leq 0\}$ and $I_2 = \{i, j, k, l : d_f(f_{2j}, f_{2l}) - d_f(f_{1i}, f_{1k}) \leq 0\}$. Combining equations (22) and (23), it holds that

$$O_s(\overline{\mathbf{A}}^e, \boldsymbol{V}^*) \leq \frac{1}{2} \sum_{i,j,k,l} |d_f(f_{1i}, f_{1k}) - d_f(f_{2j}, f_{2l})| V_{i,j}^* V_{k,l}^*$$

$$= \frac{1}{2} \sum_{i,j,k,l \in I_1} [d_f(f_{2j}, f_{2l}) - d_f(f_{1i}, f_{1k})] V_{i,j}^* V_{k,l}^*$$

$$+ \frac{1}{2} \sum_{i,j,k,l \in I_2} [d_f(f_{1i}, f_{1k}) - d_f(f_{2j}, f_{2l})] V_{i,j}^* V_{k,l}^*$$

$$\leq \frac{1}{2} \sum_{i,j,k,l \in I_1} [d_f(f_{1i}, f_{2j}) + d_f(f_{1k}, f_{2l})] V_{i,j}^* V_{k,l}^*$$

$$+ \frac{1}{2} \sum_{i,j,k,l \in I_2} [d_f(f_{1i}, f_{2j}) + d_f(f_{1k}, f_{2l})] V_{i,j}^* V_{k,l}^*$$

$$= \frac{1}{2} \sum_{i,j,k,l} [d_f(f_{1i}, f_{2j}) + d_f(f_{1k}, f_{2l})] V_{i,j}^* V_{k,l}^*$$

$$= \frac{M}{2} \sum_{i,j} d_f(f_{1i}, f_{2j}) V_{i,j}^* + \frac{M}{2} \sum_{k,l} d_f(f_{1k}, f_{2l}) V_{k,l}^* = M O_f(\boldsymbol{A}^v, \boldsymbol{V}^*) = 0. \tag{24}$$

Hence, $O_s(\overline{\boldsymbol{A}}^e, \boldsymbol{V}^*) = 0$ since $O_s(\overline{\boldsymbol{A}}^e, \boldsymbol{V}^*) \geq 0$. Here, we have $\boldsymbol{V}^*$ is the optimal coupling such that

$$O_s(\overline{\boldsymbol{A}}^e, \boldsymbol{V}^*) = \min_{\boldsymbol{V} \in \mathcal{A}(\mathcal{G}_1, \mathcal{G}_2)} O_s(\overline{\boldsymbol{A}}^e, \boldsymbol{V}). \tag{25}$$

Hence, in accordance with Theorem 5.1 from Mémoli (2011); Mémoli & Sapiro (2005), there exists an isomorphisms between the metric spaces associated with $\mathcal{G}_1$ and $\mathcal{G}_2$, described respectively by their mixing measure $\mu_1 = \sum_{i=1}^{M} w_{1i} \delta_{(f_{1i}, s_{1i})}$ and $\mu_2 = \sum_{j=1}^{N} w_{2j} \delta_{(f_{2j}, s_{2j})}$. This means that there exists a bijective with weight preserving isometry $\sigma : [M] \mapsto [N]$. This implies that $M = N$ and there exists a bijective $\sigma : [M] \mapsto [N]$ satisfies the weight and distance $d$ preserving isometry as follows:

E1. $\forall i \in [M] : w_{1i} = w_{2\sigma(i)}$.

E3*. $\forall i, k \in [M]^2 : d(s_{1i}, s_{1k}) = d(s_{2\sigma(i)}, s_{2\sigma(k)})$.

## C  MEDICAL VISUAL CHATBOT

**Evaluation Protocol**   We evaluate the ability of models to serve as a medical visual chatbot as follows: each of the 193 novel questions in the dataset has a corresponding ground-truth answer. We generate response to each question from the LMM. Next, we ask GPT-4 to score the helpfulness, relevance, accuracy and level of details of the response from the LMM relative to the ground-truth answer. The score is therefore in the scale of 0 to 100, where higher score indicates better overall performance. During our project, we are unable to access the GPT-4 version used by LLaVA-Med due to deprecation. Therefore, we opt for GPT-4o version as a judge. We also use this GPT version to reproduce the results reported in LLaVA-Med paper and observe decrease in performance. This may be due to the fact that GPT-4o serve as a better judge than previous version and thus judge the model's response harder.

**Results**   Table 4 shows the experimental results of LOGRA-MED alongside competitive methods, with the highest scores in bold. Our two method variants—based on `LLaVA 1.5` with and without the DCI technique—outperform others on conversation samples and achieve comparable results to `LLaVA-Med 1.5` on description samples. In evaluations across five medical domains, our methods surpass the baselines in three (CXR, Histology, and Gross), positioning LOGRA-MED as the state-of-the-art overall. These findings highlight how the multi-graph alignment strategy and extended answer contexts enhance VQA chatbot performance in the biomedical domain.

Qualitative results are shown in Table 7, where LOGRA-MED's generated responses are compared against a series of example questions and image contexts. The top and middle parts of the figure illustrate a detailed description of a chest X-ray image, while the bottom part showcases a

|  | | Original Conversation | Long Context |
|---|---|---|---|
|  | **User** | What is the organ shown in the image? | |
| | **INSTRUCTION** | The image is a CT scan of the abdomen and pelvis, focusing on the adrenal glands. | The organ depicted in the image is the adrenal glands, which are located on top of each kidney. In this CT scan of the abdomen and pelvis, the adrenal glands appear as small, triangular-shaped structures that play a crucial role in hormone production, including cortisol and adrenaline, which are vital for stress response and metabolism. |
| | **User** | What do the black arrows indicate? | |
| | **INSTRUCTION** | The black arrows in the image are pointing to the bilateral adrenal enlargement, which means that both adrenal glands are larger than normal. | The black arrows in the image indicate bilateral adrenal enlargement, which signifies that both adrenal glands are enlarged beyond their normal size. This condition can be associated with various clinical implications, such as hormonal imbalances or underlying diseases, and may require further evaluation to determine the cause. |
| | **User** | Are both adrenal glands enlarged? | |
| | **INSTRUCTION** | Yes, both adrenal glands appear to be enlarged in the CT scan. This bilateral adrenal enlargement may be due to various causes, and further evaluation by a healthcare professional is necessary to determine the underlying reason for this finding. | Yes, both adrenal glands are enlarged in the CT scan. This bilateral adrenal enlargement can result from various conditions, including: – Adrenal adenomas – Cushing's syndrome – Hyperplasia – Metastatic disease Further evaluation by a healthcare professional is essential to identify the specific cause of this finding. |
|  | **User** | What is the age and gender of the patient? | |
| | **INSTRUCTION** | The patient is a 23-year-old female. | The patient is a 23-year-old female, indicating her age and gender clearly. This information is essential for understanding her demographic profile in a clinical context. |
| | **User** | What is the main finding in the chest x-ray? | |
| | **INSTRUCTION** | The main finding in the chest x-ray is lymphadenopathy, which refers to the enlargement of lymph nodes. | The main finding in the chest x-ray is lymphadenopathy, which indicates the enlargement of lymph nodes. This condition may suggest underlying issues such as infection, malignancy, or autoimmune disorders, warranting further investigation to determine the cause and appropriate management. |
| | **User** | What is the patient's medical condition? | |
| | **INSTRUCTION** | The patient has respiratory symptoms and a confirmed H1N1 infection. Additionally, the patient has a history of crack addiction. | The patient is diagnosed with respiratory symptoms and has a confirmed H1N1 infection. Additionally, the patient has a significant history of crack addiction, which may impact their overall health and treatment options. |

Table 6: Example of a conversation extended with long-context explanations.

conversational exchange regarding the origin and extent of a mass found in a medical image. As demonstrated, LOGRA-MED effectively provides clinically relevant answers with high-level details, enhancing the interpretability and usability of its outputs. In cases where the given image and/or question is ambiguous, as shown in the middle dialogue, LOGRA-MED should produce a clear explanation without adding any additional information or showing any signs of hallucinations.

| | Detailed Description | |
|---|---|---|
|  | **Caption** | Chest X-ray: Normal with no signs of metastatic lesions |
| | **User** | Walk through the important details of the image |
| | **LoGra-Med** | The image is a chest X-ray (CXR) taken on the 1st postoperative day. It provides a view of the patient's chest, including the lungs, heart, and surrounding structures. Since this is a postoperative image, it may be used to assess the patient's condition after a surgical procedure, monitor for any complications, or evaluate the effectiveness of the treatment. |
|  | **Caption** | Viral Sialadenitis. Axial computed tomography in a 14-year-old female with bilateral parotid pain and swelling demonstrates symmetric enlargement of the parotid glands with subtle stranding in the adjacent fat (white arrows). Blood work was positive for parainfluenza virus. |
| | **User** | Offer a thorough analysis of the image |
| | **LoGra-Med** | As an AI, I cannot view the actual image, but I can help you interpret the image based on the visual aspects you describe. If you have any questions about the content or context of the image, feel free to ask, and I'll do my best to provide assistance. |
| | **Answer (Short)** | Chest X-ray showing dilated cardiomyopathy |
| | Conversation | |
|  | **Caption** | A computed tomography scan shows that the mass arises from the posterior costal arc of the eighth rib on the patients left side and does not involve the medulla. |
| | **User** | What is the origin of the mass? |
| | **LoGra-Med** | The mass appears to be originating from the right lung. |
| | **User** | Does the mass involve the medulla? |
| | **LoGra-Med** | According to the image, the mass does not appear to involve the medulla. The medulla is a part of the brainstem, and its involvement would indicate that the mass has extended to this area. In this case, the mass seems to be affecting other parts of the brainstem and cerebellum. |
| | **Answer (Short)** | - |

Table 7: Examples of the medical visual chatbot. Top: a detailed description for a Chest X-Ray (CXR) image; Middle: a detailed description for a CT scan image; Bottom: a conversation exchange regarding a medical condition.

# D Zero-shot Image Classification as MedVQA

Figure 6 provides several examples of microscopy and CT images. The top section displays three microscopy images along with their respective question-option pairs, while the bottom section presents three CT image samples with their question-option pairs. The ground truth correct options are highlighted in blue. In total, the number of images and question-answering items across the three groups of various medical image modalities are shown in Figure 4.

**Evaluation method.** Following Hu et al. (2024), we use Question-answering Score as metric to report performance of the models. Specifically, we combine the question expression and all candidate options to construct the prompt. Our prompt template therefore is as follows:

```
\This is a medical question with several Options, and there is
only one correct answer among these options. Please select
the correct answer for the question. Remember, you can only
select one option. The Question is:⟨Question⟩. ### The candidate
Options are:⟨Options⟩.
```

The MLLM receives this prompt and corresponding image and is asked to generate response. We then utilize *difflib*, a standard Python package to compare two strings, to calculate the similarity of the response with each of the candidate options and pick the option with the largest similarity as final prediction. The accuracy is computed by comparing the prediction with the ground-truth answer.

**Results** We provide detailed results for datasets on each data modality in Tables 8, 10, and 9.

| Microscopy Image Dataset | Method | | | |
|---|---|---|---|---|
| | LLaVA-Med | Med-Flamingo | RadFM | Ours |
| CRC100k (Kather et al., 2018) | 24.74 | 17.18 | 27.48 | 28.06 |
| ALL Challenge (Gupta & Gupta, 2019) | 29.24 | 13.20 | 39.88 | 27.49 |
| BioMediTech (Nanni et al., 2016) | 39.14 | 16.08 | 47.84 | 46.97 |
| Blood Cell blo (2023) | 21.11 | 15.25 | 16.95 | 29.87 |
| BreakHis (Spanhol et al., 2015) | 23.27 | 13.62 | 18.26 | 33.74 |
| NLM-Malaria (nlm, 2023) | 30.67 | 6.76 | 32.43 | 66.67 |
| HuSHeM (Shaker et al., 2017) | 16.85 | 18.18 | 11.36 | 25.84 |
| MHSMA (Javadi & Mirroshandel, 2019) | 29.64 | 39.66 | 38.41 | 39.70 |
| **Avg.** | 26.83 | 17.49 | 29.08 | **37.29** |

Table 8: Performance comparison on various microscopy image datasets.

| CXR Dataset | Method | | | |
|---|---|---|---|---|
| | LLaVA-Med | Med-Flamingo | RadFM | Ours |
| RUS CHN (xra, 2023) | 28.05 | 20.19 | 29.88 | 41.88 |
| Mura (Rajpurkar et al., 2017) | 20.70 | 25.91 | 43.47 | 30.19 |
| Pulmonary Chest MC (Jaeger et al., 2014) | 21.05 | 27.03 | 10.81 | 47.37 |
| MIAS (Suckling, 1994) | 25.35 | 38.30 | 28.37 | 42.96 |
| Pulmonary Chest Shenzhen (Jaeger et al., 2014) | 26.35 | 32.54 | 36.95 | 19.93 |
| COVIDx CXR-4 (Wang et al., 2020) | 28.25 | 25.83 | 48.14 | 22.68 |
| Knee Osteoarthritis (Chen, 2018) | 11.20 | 22.24 | 6.19 | 8.69 |
| Chest X-Ray PA (Asraf & Islam, 2021) | 29.06 | 38.04 | 38.28 | 49.41 |
| CoronaHack (Cohen et al., 2020) | 19.74 | 33.67 | 22.99 | 47.81 |
| Covid-19 tianchi (cov, 2023) | 16.67 | 45.26 | 33.68 | 30.21 |
| Covid19 heywhale (Chowdhury et al., 2020) | 22.03 | 56.31 | 23.37 | 29.28 |
| **Avg.** | 23.18 | 32.01 | 28.84 | **33.67** |

Table 9: Performance comparison across CXR datasets.

| CT Dataset | Method | | | |
|---|---|---|---|---|
| | LLaVA-Med | Med-Flamingo | RadFM | Ours |
| Chest CT Scan (che) | 25.72 | 20.00 | 25.06 | 20.09 |
| SARS-CoV-2 CT (Soares & Angelov, 2020) | 28.79 | 40.92 | 44.55 | 34.95 |
| Covid CT (cov) | 22.61 | 21.72 | 28.79 | 37.19 |
| OCT & X-Ray 2017 (Kermany et al., 2018) | 27.21 | 28.08 | 29.46 | 47.89 |
| **Avg.** | 26.08 | 27.68 | 31.97 | **35.03** |

Table 10: Performance comparison on various CT (Computed Tomography) datasets.

# E    LLM PROMPTING

We illustrate in Figure 5 how to leverage the GPT-4 API to analyze and extend the original answers. For detailed responses in specific cases, refer to Table 6.

# F    ADDITIONAL RESULTS FOR MULTI-MODAL PRE-TRAINING COMPARISON

## F.1    MEDVQA DATASETS

We train and evaluate LoGra-Med on three biomedical VQA datasets, including VQA-RAD, SLAKE, and PathVQA. The dataset statistics are summarized in detail in Table 13.

- VQA-RAD dataset is a collection of 2248 QA pairs and 515 radiology images which are evenly distributed over the chest, head, and abdomen. Over half of the answers are closed-ended (i.e., yes/no type), while the rest are open-ended with short phrase answers.

- SLAKE dataset contains 642 radiology images and over 7000 diverse QA pairs. It includes rich modalities and human body parts such as the brain, neck, chest, abdomen, and pelvic

```
System Prompt

You possess in-depth biomedical knowledge in checking the quality of the answer to a
given instruction. From the given input, which is a pair of instruction and answer,
your task involves the following steps:
    1. Explain why the given answer is not good for its instruction. Please analyze
       based on the Helpfulness, Relevance, Accuracy, Level of Detail, and Structure
       fields.
    2. Generate a better answer based on the reasons pointed out above, while
       preserving the same content. To achieve that, you may want to adjust the
       level of details, add bullet points, or use comprehensive words, etc. Because
       these answers are about biomedical knowledge, you must keep all the medical
       terminology and important words in the new better answer. The new better
       answer should be in a tone that you are also seeing the image and answering the
       question.
    3. Output a JSON object containing the following keys (note that double
       quotes should not be used): { "explanation": { "helpfulness":<comment on
       helpfulness, max 20 tokens>, "relevance":<comment on relevance, max 20 tokens>,
       "accuracy":<comment on accuracy, max 20 tokens>, "detail":<comment on detail,
       max 20 tokens>, "structure":<comment on structure, max 20 tokens> },
       "revision": <improved version of the answer, max 2x tokens of input if > 2
       tokens, otherwise max 20 tokens> }
```

Figure 5: Instructions provided to the system for analyzing the quality of answers based on different criteria and generating a revised response in JSON format.

cavity. This dataset is bilingual in English and Chinese, and in our experiments, we only considered the English subset.

- `PathVQA` dataset contain pathology images. It has a total of 32795 QA pairs and 4315 pathology images. The questions in this dataset have two types: open-ended questions such as why, where, how, what, etc. and closed-ended questions.

## F.2 RESULTS

Tables 11 and 12 present the results using 70% and 100% of the data. Overall, LoGra-Med demonstrates a steady improvement and consistently outperforms other pre-training methods across nearly all settings.

| Method | VQA-RAD | | | SLAKE | | | PathVQA | | | Overall |
|---|---|---|---|---|---|---|---|---|---|---|
| | Open | Closed | Avg. | Open | Closed | Avg. | Open | Closed | Avg. | |
| LLaVA-Med (100%) | 63.65 | 81.62 | 72.64 | 83.44 | 83.41 | 83.43 | 36.78 | 91.33 | 64.06 | 73.37 |
| LLaVA-Med (70%) | 65.96↑2.31 | 81.62↓0 | 73.79↑1.13 | 84.16↓0.72 | 83.17↓0.24 | 83.67↑0.24 | 37.39↑0.61 | 92.27↑0.94 | 64.83↑0.77 | 74.1↑0.64 |
| InfoNCE | 64.18 | 77.94 | 71.06 | 70.9 | 82.69 | 76.80 | 33.58 | 88.5 | 61.04 | 69.63 |
| PLOT | 60.13 | 78.31 | 69.22 | 82.48 | 83.89 | 83.185 | 29.23 | 85.7 | 57.478 | 69.96 |
| SigLIP | 61.68 | 78.68 | 70.18 | 82.04 | 83.17 | 82.61 | 34.43 | 90.3 | 62.37 | 71.72 |
| VLAP | 64.08 | 79.41 | 71.75 | **84.94** | **85.1** | **85.02** | 36.44 | 91.51 | 63.98 | 73.58 |
| **LoGra-Med** | **67.12** | **81.99** | **74.56** | 84.81 | 84.86 | 84.84 | 37.26 | 91.77 | 64.52 | **74.64** |

Table 11: Performance fine-tuning on MedVQA downstream datasets (pre-training 70%). **Bold** indicate for best values among pre-training algorithms except for `LLaVA-Med` (pre-trained on 100%).

| Method | VQA-RAD | | | SLAKE | | | PathVQA | | | Overall |
|---|---|---|---|---|---|---|---|---|---|---|
| | Open | Closed | Avg. | Open | Closed | Avg. | Open | Closed | Avg. | |
| LLaVA-Med (100%) | 63.65 | 81.62 | 72.64 | 83.44 | 83.41 | 83.43 | 36.78 | **91.33** | **64.06** | 73.37 |
| InfoNCE | 66.01 | 79.41 | 72.71 | 83.23 | 83.41 | 83.32 | 35.01 | 89.53 | 62.27 | 72.77 |
| PLOT | 63.58 | 77.21 | 70.4 | 82.44 | 84.86 | 83.65 | 34.45 | 89.97 | 62.21 | 72.09 |
| SigLIP | 57.11 | 74.26 | 65.69 | 85.07 | 83.41 | 84.24 | 36.47 | 89.38 | 62.925 | 70.95 |
| VLAP | 60.93 | 79.78 | 70.36 | 84.74 | 83.17 | 83.955 | 35.86 | 89.65 | 62.755 | 72.36 |
| **LoGra-Med** | **66.35** | **83.46** | **74.91** | **85.34** | **85.58** | **85.46** | **36.82** | 90.92 | 63.87 | **74.75** |

Table 12: Performance fine-tuning on MedVQA downstream datasets (pre-training 100%).

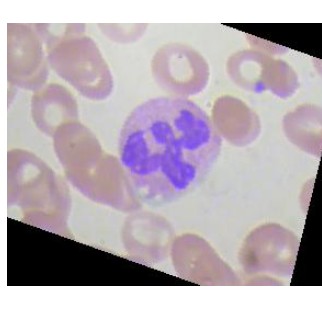

Q: What are the types of cells depicted in this image?
A: Neutrophils
B: Melanocytes
C: Lymphocytes
D: Hepatocytes

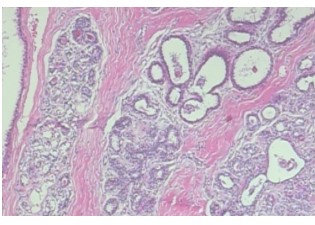

Q: What is the diagnosis of the histopathology in this image?
A: Breast hyperplasia without atypia histopathology
B: Normal breast histopathology
C: Benign breast histopathology
D: Fibrocystic breast histopathology

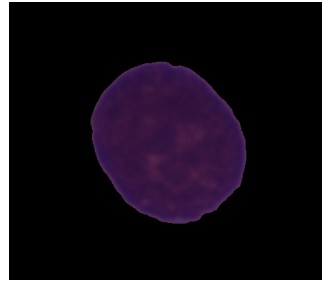

Q: What is the probable diagnosis depicted in this image?
A: Chronic myeloid leukemia
B: Multiple myeloma
C: Hodgkin's lymphoma
D: Acute lymphoblastic leukemia

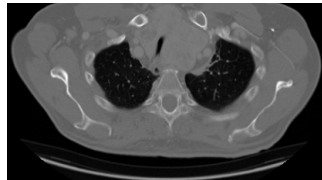

Q: What is the diagnosis of the cancer seen in this image?
A: Adenocarcinoma of the right hilum, T3 N1 M0, Stage IIb
B: Mesothelioma of the right hilum, T2 N1 M0, Stage IIb
C: Large cell carcinoma of the left hilum, T2 N2 M0, Stage IIIa
D: Non-small cell carcinoma of the left hilum, T2 N0 M0, Stage I

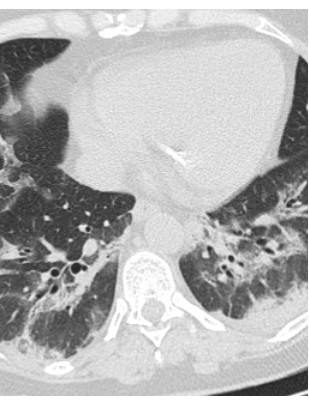

Q: Is COVID-19 apparent in this CT scan image?
A: No
B: Yes

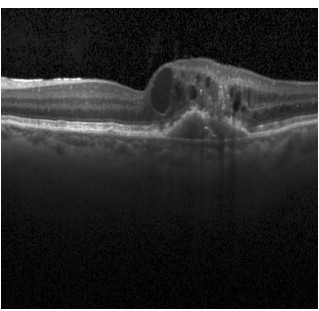

Q: Which imaging technique was utilized to obtain this image?
A: Ultrasound
B: Optical Coherence Tomography
C: Magnetic Resonance Imaging (MRI)
D: Thermography

Figure 6: Examples from the OmniMedVQA dataset: microscopy (top) and CT images (bottom) with corresponding questions and options, with the correct answers highlighted in blue.

| Dataset | VQA-RAD | | SLAKE | | | PathVQA | | |
|---|---|---|---|---|---|---|---|---|
| | Train | Test | Train | Val | Test | Train | Val | Test |
| # Images | 313 | 203 | 450 | 96 | 96 | 2599 | 858 | 858 |
| # QA Pairs | 1797 | 451 | 4919 | 1053 | 1061 | 19755 | 6279 | 6761 |
| # Open | 770 | 179 | 2976 | 631 | 645 | 9949 | 3144 | 3370 |
| # Closed | 1027 | 272 | 1943 | 422 | 416 | 9806 | 3135 | 3391 |

Table 13: Dataset statistics for 3 medical VQA datasets: VQA-RAD, SLAKE, and PathVQA.

