# OpenReview forum: "LoGra-Med: Long-Context Multi-Graph Alignment for Medical Visual-Language Models"
_ICLR.cc/2025/Conference — ICLR 2025 Conference Withdrawn Submission_

### Official Review · Reviewer_cEx3 · 2024-10-27

**Soundness:** 3
**Presentation:** 2
**Contribution:** 2
**Rating:** 5
**Confidence:** 4

**Summary:**

This paper introduces LOGRA-MED, a novel approach addressing the data efficiency challenge in medical multi-modal large language models (med-MLLM) through multi-graph alignment. The work reveals that current autoregressive training methods like LLAVA-Med require extensive instruction-following data, showing a significant performance drop from 72.64% to 52.39% on VQA-RAD when reducing training data to 10%. To address this, LOGRA-MED enforces triplet correlations in the latent space among image features, instruction data, and GPT-4 generated extended contextual captions using a structure-aware graph matching framework. The method efficiently handles the combinatorial complexity of multi-graph alignment through an implicit maximum likelihood estimation approach with black-box gradient computation.

**Strengths:**

The technical innovation of LOGRA-MED lies in its formulation of vision-language alignment as a multi-graph matching problem, generalizing beyond traditional pairwise contrastive learning approaches. The theoretical foundation is particularly strong, proving both metric and geodesic properties of the graph alignment distance. The implementation cleverly circumvents the NP-hard nature of multi-graph matching using Lagrange decomposition and efficient heuristic solvers. The empirical results are comprehensive, demonstrating that with only 10% of training data, LOGRA-MED achieves 72.52% accuracy on VQA-RAD, nearly matching LLAVA-Med's 72.64% with full data. The method's effectiveness extends across various tasks including medical visual chatbot capabilities (44.82% overall score) and zero-shot classification on 23 datasets spanning microscopy, CT, and CXR modalities.

**Weaknesses:**

The reliance on GPT-4 for generating extended contexts raises questions about potential biases and the method's generalizability. While the paper demonstrates improved performance, the computational complexity analysis could be more thorough - the reported one-hour additional training time compared to LLAVA-Med merits deeper discussion given the sophisticated graph matching operations. The variable performance across different medical domains (particularly in PathVQA and some zero-shot classification tasks) suggests potential limitations in handling diverse medical contexts. The ablation studies, while informative, could benefit from exploring different graph construction strategies beyond k-nearest neighbors and analyzing the impact of various distance metrics in the graph matching formulation.

**Questions:**

The sensitivity of LOGRA-MED to the quality and consistency of GPT-4 generated contexts remains unclear？

how would the method perform with simpler context extension approaches or with different language models?

The paper could elaborate on the interplay between the traditional autoregressive loss and the graph alignment objective, particularly regarding their relative contributions to the final performance？

The choice of hyperparameters in the graph construction phase, especially the selection of k in k-nearest neighbors, appears critical but lacks detailed justification？

---

> ### Author Response · Authors · 2024-11-23
> **Response to Reviewer cEx3**
>
> Dear Reviewer cEx3, we appreciate your positive feedback on our proposed algorithm. Below, we address your remaining concerns regarding the stability of LoGra-Med with respect to paraphrased text generated by other LLM models, as well as other questions you raised.
>
> **Q1. The potential sensitivity LoGra-Med on GPT-4, how does it work with other language models or simpler context extension?**
>
> To validate the robustness of LoGra-Med w.r.t the LLMs used to generate extended captions, we conducted additional experiments. (i) First, we perform simple paraphrasing during instruction tuning by replacing keywords with their synonyms, and (ii) We use the Gemini API from Google to generate paraphrase captions.
>
> We test (i) on weights pre-training with 40% data and (ii) for both 10% and 40% pre-training data. Model performance are evaluated on VQA-RAD and SLAKE, with average scores computed for open- and closed-based questions in these datasets. These new experiments are reported in **Table 7** in the attached link https://anonymous.4open.science/r/logra-med-386F/readme.md.
>
> We conclude that LoGra-Med continues to perform well when using Gemini LLM models, achieving significantly better results than the LLaVa-Med baseline, especially on the 10% and 40% data subsets of VQA-RAD. In contrast, a simpler model version does not produce similar improvements. This difference can be attributed to the lack of richer, more contextually aware descriptions incorporated into LoGra-Med, which helps the model better use the image-caption relationships.
>
> **Q2. Variable performance on fine-tuned PathVQA and Zero-shot classification task.**
>
> We believe that the variations in LoGra-Med’s performance can be attributed to task-specific adaptation. In the case of PathVQA, the model was fine-tuned with auto-regressive learning on the provided training samples, allowing it to specialize its general knowledge for the domain-specific task. In contrast, the zero-shot image classification task does not benefit from fine-tuning, which makes it more difficult for the model to generate accurate predictions, particularly when it hasn’t seen sufficient examples or when visual features aren’t strongly tied to the classes in the model’s knowledge base.
>
> In conclusion, we argue that the zero-shot performance of LoGra-Med and other baselines could be significantly improved by fine-tuning task-specific training data.
>
> **Q3. Analysis of graph construction step with different K values in K-nearest neighbors.**
>
> We conducted new experiments, detailed in **Table 6** (attached link: https://anonymous.4open.science/r/logra-med-386F/readme.md), to evaluate different K values used in the graph construction step. The table reports the model performance on the VQA-RAD dataset as well as the training time (Step-2 in pre-training) for each value of K. Our results show that K = 5 is the optimal choice. These findings will be included in the revised version of the paper.
>
> **Q4. Analysis of interpolation between the traditional autoregressive loss and the graph alignment objective.**
>
> In our default configuration, we choose alpha = 1 as the coefficients to combine auto-regressive and graph alignment objectives. To verify this choice, we ran further another ablation study with alpha = 0.1, 0.5 and trained LoGra-Med on 10% pre-training data. We present results on the VQA-RAD and Slake datasets in **Table 8, Section F** (attached link: https://anonymous.4open.science/r/logra-med-386F/readme.md).
>
> The records demonstrate that alpha = 1.0 is the optimal choice; when reducing the contribution of multi-graph alignment, the model tends to drop their results.
>
> **Q5. Analysis of Extra Running Time of LoGra-Med (1 hour)**
>
> We attribute the increased training time to two main factors: (a) solving three graph alignment problems simultaneously and (b) the need to compute second-order graph alignments using heuristic solvers and perform backward gradient calculations with black-box gradient estimations.
>
> Despite the additional training time required by LoGra-Med, this extra cost leads to better performance on downstream tasks. Furthermore, an ablation study showed that even when LLaVa-Med is trained with or without extended contexts and for a longer duration, our model consistently outperforms LLaVa-Med.

---

> > ### Author Response · Authors · 2024-11-24
> > **Look forward to your response**
> >
> > Dear Reviewer cEx3,
> >
> > We would like to thank you very much for insightful review, and we hope that our response addresses your previous concerns regarding our paper. However, as the discussion period is expected to end in the next few days, please feel free to let us know if you have any further comments on our work. We would be willing to address any additional concerns from you. Otherwise, we hope that you will consider increasing your rating.
> >
> > Thank you again for spending time on the paper, we really appreciate it!
> >
> > Best regards,
> >
> > Authors

---

> > > ### Author Response · Authors · 2024-11-25
> > > **Look forward to your response (kindly remind)**
> > >
> > > Dear Reviewer cEx3,
> > >
> > > We hope this message finds you well. As the discussion deadline is fast approaching, we wanted to kindly remind you that we have not yet received your feedback.
> > >
> > > In our rebuttal, **we conducted four additional experiments (detailed in Tables 6, 7, and 8: https://anonymous.4open.science/r/logra-med-386F/readme.md)**, including an evaluation **using the Gemini API** to validate the robustness of LoGra-Med with other LLM models. This particular experiment required significant effort and budget due to the costs associated with querying the API. Therefore, we would greatly appreciate it if you could take the time to review our responses, supported by these new results before the discussion period concludes.
> > >
> > > Thank you very much, and we look forward to hearing from you.
> > >
> > > Best regards,
> > > The Authors

---

> ### Author Response · Authors · 2024-12-01
> **Look forward to your response (2 days left)**
>
> Dear Reviewer cEx3,
>
> We appreciate your efforts in reviewing our paper. We hope that our response and revision can address your concerns. There are only two days left until the end of the discussion period. We look forward to hearing from you. Please feel free to leave any additional questions or comments. Otherwise, we would greatly appreciate it if you could raise your rating.
>
> Thank you very much for your time and consideration!
>
> Best regards,
>
> Authors

---

> > ### Comment · Reviewer_cEx3 · 2024-12-01
> >
> > I greatly appreciate the authors' rebuttal and response, but I still maintain my score.

---

> ### Author Response · Authors · 2024-12-01
> **Response to Reviewer cEx3**
>
> Dear Reviewer cEx3,
>
> Thank you for your response. While we respect your evaluation, we kindly request further clarification on whether there are remaining issues or if additional improvements are needed, **which makes you keep the unchanged scores without further justifications**.
>
> To date, we have thoroughly addressed all your concerns through several additional experiments, as detailed in Tables 6, 7, and 8 in this link (https://anonymous.4open.science/r/logra-med-386F/readme.md). Furthermore, we extended our evaluation by generating new results for LoGra-Med using the Gemini API, creating extended captions for 24K conversations. This effort demonstrates that LoGra-Med performs robustly with other LLM models. Notably, this process required substantial computational resources and incurred significant costs (to call API) for pre-training and fine-tuning downstream tasks, **underscoring our commitment to addressing your feedback comprehensively**.
>
> We hope to hear your feedback
>
> Regards
>
> Authors

---

### Official Review · Reviewer_9ZPV · 2024-11-03

**Soundness:** 3
**Presentation:** 3
**Contribution:** 2
**Rating:** 5
**Confidence:** 2

**Summary:**

In this paper, the authors introduced a self-supervised multi-graph alignment objective strategy for pre-training medical MLLM. Specifically, the proposed method aligns the triplet consisting of the input image, its instruction data, and its extended long context generated by GPT. The graphs of images, the instruction data, and the long context are trained to align with the barycenter graph. The authors perform experiments on multiple downstream tasks, and the proposed method outperforms baseline methods. The proposed method achieved better performance with the limited amount of pre-training dataset.

**Strengths:**

1. The empirical results demonstrate that the proposed approach can reduce the demand for pre-training data for MLLM alignment, outperforming previous approaches.

2. The authors conducted a set of ablation studies, which demonstrated the contribution of each component.

3. The visualization in the manuscripts helped me to better understand the proposed approach and experiments.

**Weaknesses:**

1. From Table 5, it seems that the generated long context data plays an important role in LoGra-Med. Without it, the performance is slightly lower than the baseline LLaVA-Med. Since LLaVA-Med doesn't use the generated long context data, these results cast doubt on the contribution of the graph alignment algorithm. For a fair comparison to validate that the performance gain is not mainly due to the extra data, the author could also pre-train LLaVA-Med with the long context data and compare the performance.

2. The proposed method's performance gain may come from the extra computing in training (the authors mention that the training time is longer than the baseline method by 1 hour) rather than the method itself. For a fair comparison, the authors may pre-train the LLaVA-Med baseline method using some self-supervised objectives (e.g., InfoNCE) or additional long context data for 1 hour.

3. The authors need to mention some of the important hyperparameters used. For example, what's the batch size B? I would assume the larger the batch size, the larger the graph. And what's the value of k used in the k-NN algorithm for building edges?

**Questions:**

1. Can you provide the hyperparameters used in the paper, including the batch size B and the value of k used in the k-NN?

2. Are there any experiment results presented in the paper that can prove the performance gain compared to LLaVA-Med is due to the graph alignment algorithm itself rather than additional data or extra computing?

---

> ### Author Response · Authors · 2024-11-23
> **Response to Reviewer 9ZPV**
>
> Dear Reviewer 9ZPV,
>
> First, we sincerely thank you for taking the time to provide us with your valuable feedback. Below, we address your concerns in detail.
>
> **Q1. Performance comparison with the LLaVa-Med trained with extended context and longer training time.**
>
> We composed this question based on two first concerns and the second question of the Reviewer (Weaknesses and Question section). To address this, we conducted the following experiments:
>
> + We train LLaVa-Med baseline with **10%** pre-training data (both Stages) using **extended captions in their auto-regressive training loss**.
>
> + We train  LLaVa-Med baseline with **40%** pre-training data (both Stages) using **extended captions in their auto-regressive training loss**.
>
> Both settings were trained for 3 epochs, with training times of 0.47 and 1.9 hours, respectively. In comparison, we trained LoGra-Med for 2 epochs, ensuring that LLaVa-Med had a longer training duration than ours (our training times were 0.45 and 1.8 hours). Afterward, we fine-tuned the pre-trained models on downstream tasks. The results are summarized in **Tables 4 and 5** in the attachment link https://anonymous.4open.science/r/logra-med-386F/readme.md (highlighted in grey color).
>
> We observe that incorporating extended captions during the pre-training phase of LLaVa-Med enhances its performance compared to the default settings. However, both of these new versions still fall short of LoGra-Med’s performance. This suggests that our graph alignment algorithm plays a significant role in enhancing LoGra-Med's performance.
>
> **Q2.  Provide more details for hyper-parameters like batch size B, and values of k in graph construction.**
>
> We apologize for the missing report on these parameters. We used a batch size of  B = *16* x *number of GPUs*, and set k = 5  for graph construction. To validate this choice of k , we present an additional ablation study on the VQA-RAD dataset, as shown in **Table 6** in the attached link https://anonymous.4open.science/r/logra-med-386F/readme.md. The results indicate that k = 5  strikes the optimal balance between performance and complexity. We will add these new results and provide more details for hyper-parameters in the revised version.

---

> > ### Author Response · Authors · 2024-11-24
> > **Look forward to your response**
> >
> > Dear Reviewer 9ZPV,
> >
> > We would like to thank you very much for insightful review, and we hope that our response addresses your previous concerns regarding our paper. However, as the discussion period is expected to end in the next few days, please feel free to let us know if you have any further comments on our work. We would be willing to address any additional concerns from you. Otherwise, we hope that you will consider increasing your rating.
> >
> > Thank you again for spending time on the paper, we really appreciate it!
> >
> > Best regards,
> >
> > Authors

---

> > > ### Author Response · Authors · 2024-11-25
> > > **Looking for your feedback when discussion will be ended soon**
> > >
> > > Dear Reviewer 9ZPV,
> > >
> > > We hope this message finds you well. **As the discussion deadline approaches and we have not yet received your feedback** since posting our response, we kindly ask if there are any additional concerns we may have overlooked. In our rebuttal, we addressed your key points, including:
> > >
> > > - The performance of LLaVa-Med trained with extended contexts as compared to LoGra-Med,
> > > - A performance comparison after extending the training duration for LLaVa-Med and
> > > - Additional hyper-parameter insights supported by extra ablation studies.
> > >
> > > We greatly appreciate your time and thoughtful review.
> > >
> > > Best regards,
> > > The Authors

---

> ### Author Response · Authors · 2024-12-01
> **Clarification on your decision**
>
> Dear Reviewer 9ZPV,
>
> Thank you for providing your feedback. **We respectfully seek further clarification regarding your decision to maintain the same score, as this was done without additional explanation despite our comprehensive efforts to address your concerns**. Could you kindly indicate whether additional studies or improvements are required, or if the results provided still fall short of your expectations?
>
> We want to summarize again your concerns addressed during the rebuttal:
> - We provided the hyperparameters used in our method and conducted extra experiments to validate these choices (Tables 4 and 5 in https://anonymous.4open.science/r/logra-med-386F/readme.md).
> - We performed a direct comparison with LLaVa-Med under identical extended-context conditions and trained our method for longer durations to ensure a fair and thorough evaluation (Table 6).
>
> We look forward to hearing your thoughts.
>
> Best regards,
>
> The Authors

---

### Official Review · Reviewer_jqRH · 2024-11-04

**Soundness:** 3
**Presentation:** 2
**Contribution:** 3
**Rating:** 5
**Confidence:** 4

**Summary:**

This work propose a method that aiming for decrease the data needed for medical Multi-Modal large langauge model pre-training by enforcing triplet correlations on the latent embedding among image, descriptions, and captions. How to efficiently pre-train large models with limited data in medical domain is a quite important topic, and thus the motivation is good enough. However, the reason behind applying multi-gprah alignment is not well-explained. And this process required retrain the LLM model, which is more costful compared other methods.

**Strengths:**

1. This model propose a method to convert auto-gressive finetuning to Graph-alignment task, a fresh perspective. This seems interesting but lack of motivation.
2. The results showing that the proposed method is definetly increase the performance on different tasks with limited datasize. This is quite important and might showing a new way except data curation.
3. Some of the detail are not well-explained in the paper, please refer to the weakness and quesiton section. I will consider adjust my scores based on the authors response.

**Weaknesses:**

1. The motivation of adding graph alignment task in the pre-training stage is not explained.
2. This paper also cite the work of MedTrinity(Xie et al.2024), but you didn't include their results in your comparison tables for your experiments. Their model's performance on some dataset are better than yours, you should include them to give readers a fair comparision. You may argue that their training data is larger than yours. But your method also introduce augmented description data generated by GPT-4.
3. SInce you introduced more data by extending the short answer to long answers. So, when you finetuning the LLavaMed basedline with 10% data, do you also include the extended version answer?

**Questions:**

1. Why extending the short answer to longer-context, why would that beneficial the pre-training?
2. Have you run any qualification test on the GPT generated text?
3. Would you mind to clarify, when you mentioned "10% data", you mean, "10% datasize for the stage2 instruction-tuning stage, but the datasize for the stage1 pre-trianing is still fullzie". Is that correct?

---

> ### Author Response · Authors · 2024-11-23
> **Response to Reviewer jqRH - Part 1**
>
> Dear Reviewer jqRH, we appreciate your taking the time to give us feedback. Below, we address your concern ​​individually.
>
> **Q1. Explaining the motivation for adding graph alignment task in the pre-training stage.**
>
> We address this question through the following points:
>
> First, we explain **why doing a contrastive alignment in the pre-training step can help LLM models**. Specifically, by applying triplet contrastive learning to input images, original captions, and extended captions, the **model learns to identify relevant similarities and differences within the data rather than memorizing individual samples**. This approach thus enhances robustness, particularly when training on small pre-training datasets. Additionally, including extended captions enriches the model’s contextual and hierarchical understanding, which expands the latent space and reduces representation collapse, enabling the model to better distinguish subtle variations in medical images and descriptions.
>
> Second, among various contrastive learning approaches, we adopt a graph-based formulation inspired by prior work (LVM-Med, Duy et al., 2024). This formulation extends beyond pairwise contrastive learning by utilizing combinatorial graph matching, which captures stronger structural constraints within feature embeddings, leading to improved generalization performance. While LVM-Med is designed for aligning two graphs - vision and augmented vision - we extend this approach to three graph alignments: one for the vision graph, one for the original caption-based graph, and one for the extended caption-based graph. Our intuition is that image embeddings should be aligned with both their short and extended captions, and the LLM text embeddings for both versions should exhibit a high correlation.
>
> Based on these motivations, we designed the multi-graph alignment for pre-training. If the Reviewer is still unclear at some points, we are happy for further clarification.
>
> ------
> MH Nguyen, Duy, et al. "Lvm-med: Learning large-scale self-supervised vision models for medical imaging via second-order graph matching." NeurIPS 2024
>
> **Q2. Not Including MedTrinity results in comparison**
>
> There are two primary reasons why we did not include MedTrinity in our comparison:
>
> **Reproducibility Issues**: Using the pre-trained models provided by MedTrinity, we could not reproduce results comparable to the best version reported in their paper (denoted as LLaVA-Med++ (Ours, w/)), despite following the authors’ instructions.
>
> **Incompatible Pre-training Strategy**: MedTrinity does not offer a single unified pre-trained LLM model. Instead, separate pre-trained weights are provided for each downstream task (e.g., VQA-RAD, SLAKE, and PathVQA). This is because their approach involves combining a subset of training data from each downstream task with their collected data to create task-specific pre-trained models. For instance, to train the VQA-RAD model, they combine a subset of VQA-RAD training data with their collected data, extended with region of interest (ROI) annotations. A similar strategy is applied for SLAKE and PathVQA.
>
> In contrast, our approach (and those of other methods we compare against) involves producing a single pre-trained model that can be fine-tuned across multiple downstream tasks. We argue that this fundamental difference in methodology makes MedTrinity unsuitable for direct comparison with our framework.
>
>  **Q3. (i) Clarify the “10% data” setting. Is it "10% data size for the Stage 2 instruction-tuning stage, but the data size for the Stage 1 pre-training is still full-size; (ii) Do we include extended context in fine-tuning the LLaVaMed baseline with 10%?**
>
> We address two concerns together due to their connection to our pre-training and fine-tuning setup. Below, we recap the details of our data usage and training stages:
>
> **Pre-training**: This consists of two stages:
>
> - Stage 1: We train only the MLP to align the output of vision models, keeping all other components frozen. This follows the same strategy as LLaVa-Med, using 600k image-text pairs and an auto-regressive training loss.
>
> - Stage 2: Both the MLP and LLM are trained using auto-regressive learning combined with our multi-graph alignment approach. Specifically, we use 60k conversations, as in LLaVa-Med, for auto-regressive training, while the multi-graph alignment is trained with our extended captions derived from these 60k conversations.
>
> **Fine-tuning for downstream tasks**: Using the pre-trained model from the pre-training phase, we fine-tuned the LLM weights with only auto-regressive learning, leveraging the training samples provided for each dataset.

---

> > ### Author Response · Authors · 2024-11-23
> > **Response to Reviewer jqRH - Part 2**
> >
> > In response to your questions:
> >
> > (i) The “10%” refers to using 10% of the pre-training data for both Stage 1 and Stage 2 across all baselines. For LoGra-Med, this 10% in Stage 2 additionally incorporates our extended captions.
> > (ii) As described above, the extended captions are applied only during pre-training Stage 2. For fine-tuning downstream tasks, all methods (including ours) use the same data provided for each task and update the model using auto-regressive training alone.
> >
> > We hope these responses address your concerns. If any aspects remain unclear, we are happy to provide further clarification.
> >
> > **Q4. Why extending the short answer might be beneficial in the pre-training.**
> >
> > We incorporate extended captions as part of triplet contrastive learning between the input image, the original caption, and the extended caption. This mechanism thus can offer several benefits: (i) it enables the model to associate images with richer, domain-specific descriptions that go beyond standard captions; (ii) it aligns more closely with real-world medical workflows, where doctors leverage domain knowledge, thereby enhancing multi-scale understanding by bridging local and global features, reducing ambiguity in learning; (iii) from a representation learning perspective, extended captions can help to diversify the embedding space and capture hierarchical relationships between input images and captions, potentially improving performance in complex pre-training tasks.
> >
> >
> > **Q5. Qualification test on the GPT-generated text.**
> >
> > We adopt the GPT-4 as a tool for paraphrasing image captioning due to its improved performance compared with GPT-3.5, especially in healthcare (Jin, Hye Kyung et al. 2024). During our implementations, we also randomly checked for a hundred samples and found consistency between extended context and original ones. However, to further validate the Reviewer's concerns about the quality of the generated text, we sought help from five general practitioners currently working at top public hospitals in Vietnam (for anonymity reasons, we will update their affiliations after the review process has been completed).
> >
> > In particular, we randomly chose 1000 samples in Stage 2 of pre-training, covering five data modalities: chest X-ray, CT scan, MRI, histology, and others. Each doctor is assigned a specific data modality given their expertise, including 200 image-text pairs and corresponding captions. We then build an annotation tool for them to verify data where each sample is asked with two questions (i) whether the extended caption covers the original caption; and (ii) whether new concepts appearing in extended captions are correct. For (i) and (ii), doctors can rate with five levels (from 1 to 5), each indicating an increasing level of correctness (**Section H, anonymous link**).
> >
> > Due to the short period of time in rebuttal, until now, we only have statistics for three domains, “Chest X-Ray”, “CT-scan,” and "MRI", as shown in **Section I** in the attachment.
> >
> > It can be seen that most rating scores fall between 3 and 5, with only a small number of samples rated 1 or 2, validating the overall consistency of GPT-4 outputs. While concerns may arise regarding the impact of low-scoring extended captions (rated 1 or 2) on the LLM, it’s important to note that these extended captions are utilized solely for contrastive learning during pre-training to align the model’s latent space representations. They are not used in auto-regressive training, which involves predicting target ground-truth tokens. Additionally, the model is fine-tuned with the given training samples from downstream tasks after pre-training (no extended captions are used). Thus, we argue that the presence of a small number of noisy extended captions should not significantly affect the performance of the LLM.
> >
> > ------
> > Jin, Hye Kyung et al. “Performance of ChatGPT-3.5 and GPT-4 in national licensing examinations for medicine, pharmacy, dentistry, and nursing: a systematic review and meta-analysis." BMC Medical Education 2024.

---

> > > ### Author Response · Authors · 2024-11-24
> > > **Look forward to your response**
> > >
> > > Dear Reviewer jqRH,
> > >
> > > We would like to thank you very much for insightful review, and we hope that our response addresses your previous concerns regarding our paper. However, as the discussion period is expected to end in the next few days, please feel free to let us know if you have any further comments on our work. We would be willing to address any additional concerns from you. Otherwise, we hope that you will consider increasing your rating.
> > >
> > > Thank you again for spending time on the paper, we really appreciate it!
> > >
> > > Best regards,
> > >
> > > Authors

---

> ### Author Response · Authors · 2024-11-29
> **Look forward to your response (follow up)**
>
> Dear Reviewer jqRH,
>
> We would like to thank you very much for your insightful review, and we hope that our response addresses your concerns regarding our paper. However, as the discussion period is expected to end in the next few days, please feel free to let us know if you have any further comments on our work. We would be willing to address any additional concerns you may have.
>
> Thank you again for spending time on the paper, we really appreciate it!
>
> Best regards,
>
> Authors

---

> ### Author Response · Authors · 2024-12-01
> **Look forward to your response (kindly remind)**
>
> Dear Reviewer jqRH,
>
> We appreciate your efforts in reviewing our paper. We hope that our response and revision can address your concerns. There are only two days left until the end of the discussion period. We look forward to hearing from you. Please feel free to leave any additional questions or comments. Otherwise, we would greatly appreciate it if you could raise your rating.
>
> Thank you very much for your time and consideration!
>
> Best regards,
>
> Authors

---

> > ### Comment · Reviewer_jqRH · 2024-12-02
> >
> > I still have some concern about this work's novelty. As one of the previous work using the same combinatorio method to do the contrastive learning by employed two transformation of the same image to form two graphs. However, in this paper, the main improvement is about do the same thing but emolying two transformation for the text prompts (short, long text) to form two graphs. Therefore, I won't raise my score as the motivation of the paper is simply following of the previous work.

---

> ### Author Response · Authors · 2024-12-02
> **Last day to post your response**
>
> Dear Reviewer jqRH,
>
> We would like to kindly remind you that **today is the final day of the discussion period**, and we have not yet to receive your feedback since November 23rd. We have put a large effort into addressing your concerns, such as developing an interface and conducting a user study with practicing doctors to validate the context generated by GPT-4 models (see Sections H and I in [https://anonymous.4open.science/r/logra-med-386F/readme.md]). Given the importance of your feedback to us, we would greatly appreciate it if you could take the time to share your thoughts in the remaining hours.
>
> Thank you for your time and consideration, and we look forward to hearing from you.
>
> Best regards,
>
> The Authors
>
> Authors

---

> ### Author Response · Authors · 2024-12-03
> **Response to Reviewer jqRH**
>
> Dear Reviewer jqRH,
>
> We thank you for your response. It's very regrettable that the Reviewer made comments just before the discussion deadline a few hours and raised a new concern about the novelty of LoGra-Med compared with prior work on a single domain (LVM-Med).
>
> Below, we address your concern with a comprehensive comparison of the two approaches across several key aspects. Our formulation distinguishes itself from LVM-Med in terms of its original motivation, the design of efficient solvers for multi-domain tasks, the method of graph construction, achieved performance, and the theoretical insights contributing to a deeper understanding of the algorithm. We hope this table helps you better understand our contributions.
>
> | **Comparison**                 | **LoGra-Med**                                                                                                                                                                                                                                                                   | **LVM-Med**                                                                                |
> |-----------------------------|-----------------------------------------------------------------------------------------------------------------------------------------------------------------------|-----------------------------------------|
> | **Motivation**              | **We identified, for the first time, the data-intensive nature of auto-regressive modeling in the LLaVa-Med model** (Figure 1 of the paper). This novel observation sheds light on a previously unreported phenomenon in the context of multi-modal LLM. | Designed to learn a **single modality pre-trained model (vision)** by performing contrastive learning between vision and its augmented versions. |
> |           | Building on this insight, we modify Stage 2 of the LLaVa-Med pre-training by introducing a **multi-graph alignment** approach for **multi-modal embeddings within the large language model (LLM) across vision, image captions, and extended captions** generated by advanced models like GPT-4 or Gemini. |  |
> | **Scalable Optimization Solver** |Motivated to integrate multiple modalities (e.g., K = 3 in our setting but can be extendable to voice or medical tables) into objective functions without significant computational costs.  **Pairwise solvers in LVM-Med are unsuitable** as they require computationally expensive $\binom{K}{2}$ pairwise alignments, which becomes computationally prohibitive, particularly with large-scale LLM models containing 7B parameters. | Solves pairwise graph matching using a combinatorial objective function.                    |
> |  | We propose using a barycenter graph inspired by optimal transport theory to solve triplet alignments. Unlike previous unsupervised methods (Nguyen et al., 2024), which estimate the barycenter before alignment, we directly define it using known triplet pairs across the three graphs, enabling faster computation. | |
> |  | We evaluate the **LoGra-Med solver against the LVM-Med solver** for K = 3 domains using the full pre-training dataset. Due to triplet constraints across modalities, the results demonstrate that LoGra-Med consistently delivers stronger performance across VQA tasks. **LoGra-Med Solver: 74.9 (VQA-RAD), 85.5 (SLAKE)** | **LVM-Med Solver: 73.9 (VQA-RAD), 84.3 (SLAKE)**|
> | **Graph Construction** | We work in the vision-language domain; therefore, augmenting the input images is not straightforward because the caption no longer precisely describes the images. | Two graphs are used: the first is based on the input images, and the second represents their augmented versions.                   |
> | | We pay attention to transforming image captioning and show that (i) simply paraphrasing the caption with synonyms does not help significantly and (ii) using extended contexts powered by frozen LLM models like GPT-4 or Gemini is a solution to enhance model capacities. It’s important to note that this is the first time (up to our latest knowledge) extended contexts used instruction-following data for training LLM.  |                |
> | | Our ablation study (**Table E, bottom section, anonymous link**) confirms these findings, demonstrating the effectiveness of the **vision, captioning, and extended caption graphs** and emphasizing the role of extended context in instruction-following data -based LLM training.|                |
> | **Performance** | We do the same setting with 40% but using LoGra-Med. **Our results are 74.37 (VQA-RAD), 84.99 (SLAKE)**|We use LVM-Med as the alignment algorithm and trained with 40% pre-training data in Stage 2 of LLaVa-Med, The model is then fine-tuned on two downstream tasks: **72.12 (VQA-RAD), 81.95 (SLAKE)**|
> | **Theoretical Results** |We contribute two new theoretical insights for LoGra-Med (i) The proposed multi-graph distance is a **valid metric** (Theorem 1); (ii) **Shortest path in the geodesic space** of multi-modal graphs (Theorem 2).|None|

---

### Official Review · Reviewer_Kirs · 2024-11-04

**Soundness:** 3
**Presentation:** 3
**Contribution:** 3
**Rating:** 6
**Confidence:** 3

**Summary:**

The paper proposes a multi-graph matching objective to enhance the vision-language alignment in generative VLMs for medical tasks. The authors motivate the need for such an auxiliary learning objective from a training efficiency point of view, showing that the next token prediction loss function demands a large amount of visual instruction tuning data that's not easy to acquire. To reduce the sample complexity, this paper proposed their method of treating visual features, text features, and augmented text features (based on a more verbose version of the ground truth answer) as graphs and letting the VLM learn to match between the triplets. Since solving a multi-domain graph alignment problem is computationally expensive, the authors propose a more scalable solution by separately aligning each domains with a barycenter graph. The experiments in the paper show that multi-graph alignment significantly improves upon LlaVA-Med, at both low pretraining data (10%) and full data regimes across a wide range of tasks.

**Strengths:**

I like how the paper motivates the necessity of the proposed algorithm from the perspective of training efficiency, which is quite refreshing. The experiments also validate that the proposed algorithm indeed results in better learning outcomes at low data regimes and even does so when using all the pretraining data.

The presentation of the core methodology flows very well, with tight logical connections. The gradient-based method to by-pass the high time complexity of the solver also seems reasonable.

The paper also seems to be quite comprehensive in terms of baselines. The ablation studies are satisfactory, especially the one without “long-context”.

The experiment details in the supplemental material is very comprehensive, including important details such as system prompt and evaluation protocols.

**Weaknesses:**

I have concerns regarding the core methodology.

1. The paper seems to have skipped simpler solutions that might as well worked and jumped right into a complex proposal. In section 3.4, the authors argue the necessity of their scalable multi-graph alignment algorithm by saying pairwise graph alignment requires K choose 2 pairs. But in this specific case, K is 3 and K choose 2 is really only 3 pairs. From my point of view, this really should be the method to be tried first before moving on to more complex formulations.
2. I am too sure what leads to LoGRA’s success due to how the visual & text representations are obtained. In its essence, the representations from both modalities are sets with varying number of elements. For images, this will be different number of patches due to different resolutions; for text, it’s whatever the number of tokens the ground truth answer corresponds to. The authors are simply taking a naive average over the elements. I really doubt the average of word embeddings / visual tokens will end up giving any meaningful representations. I would like to see some analysis on this design choice, for example comparing the distribution of cosine similarity (for example) between positive and negative pairs, so that I know the proposed objective is really forcing the alignment instead of picking up some other signals.
3. As far as I know, most medical datasets are highly imbalanced, but the paper seems only to report the overall accuracy instead of per-disease macro scores. This is a bit concerning since if all the alignment algorithm does is to guild the model to favor the majority class, we will get a model with high accuracy but no practical use. I think for classification tasks, macro F1 scores should be reported. For VQA tasks, per-disease scores should be reported for datasets that include a class-label for the image-question-answer pair.
4. I think the word “long-context” is misused: from the examples, it looks like the so called “long-context” texts are no more than a few hundred words. Based on the context of the paper, the authors really meant a more verbose / extended rewrite of the original ground truth answers as an augmentation for the triplet learning. A better terminology should be used to distinguish this paper from works dealing with long-context tasks, which usually consumes at least tens of thousands of tokens for each query.

**Questions:**

1. I’m actually a bit confused on why the graph perspective is needed here. It seems the paper is solving the exactly the same problem as optimal transport, but all these optimal papers do not talk about graphs?
2. I wonder, without using the verbose version, how’s this paper different from the PLOT paper, since it’s essentially solving a matching problem between 2 domains?
3. I find counterintuitive that DCI provides mixed results. What’s the explanation here? To be honest, I would be totally fine if those “+DCI” are not there. But if the authors decide to add these experiments, then they need to explain why utilizing multi-scale features are harmful for some cases.

---

> ### Author Response · Authors · 2024-11-23
> **Response to Reviewer Kirs - Part 1**
>
> Dear Reviewer Kirs, thank you for giving us detailed feedback, especially on our methodology section, such as chosen solvers and computing features, comparing them with optimal transport, etc. In the following, we address each of your concerns with several figures and tables referred to in the anonymous link.
>
> **Q1. Testing with simpler solutions (two graph alignment and solver for multi-graph alignments).**
>
> We address this question by doing two different experiments.
>
>  (i) Contrastive loss between two graphs: one from visual features and the other is original captioning.
>
> (ii) Solving three pair-wise graph alignments in multi-graph alignment rather than solving through a barycenter graph.
>
> We provide experiments for (i) with 40% pre-training data and (ii) with 100% pre-training data. Performance is reported with Average scores on two downstream tasks (VQA-RAD and SLAKE) after fine-tuning.
>
> | Settings            | VQA-RAD | SLAKE |
> | ------------------- | ------- | ----- |
> | LoGra-Med (default), 40% | 74.37   | 84.99 |
> | LoGra-Med with (I), 40%  | 72.12   | 81.95 |
> | LoGra-Med (default), 100% | 74.91   | 85.46 |
> | LoGra-Med with (ii), 100% | 73.88   | 84.34 |
>
> The results demonstrate that incorporating extended context through multi-graph alignment significantly improves performance compared to using only two graphs, as in setting (i). Setting (ii) also highlights that solving independent pairwise graph alignments leads to lower performance than our proposed solution. This is because LoGra-Med accounts for all mutual relationships, enforcing consistency across the triplet and avoiding conflicts that may arise when pairwise alignments between subsets (e.g., A and B, B and C) contradict the relationship between A and C.
>
> To provide further insights, we also measured the running time of setting (ii) compared to the default LoGra-Med configuration. The statistics indicate that **both approaches require comparable training time**, with approximately seven hours in Stage 1 and 7.5 hours in Stage 2.
>
> In conclusion, this evidence validates the effectiveness of LoGra-Med’s configuration. Moreover, it is worth noting that our solver’s efficiency makes LoGra-Med scalable to additional domains, such as health records or surgery videos, further enhancing its applicability in building multi-modal LLMs.
>
>
> **Q2. Analysis of feature representation computing as average pooling for visual and language tokens.**
>
> We address this concern with two experiments:
>
> - We trained LoGra-Med on 70% of the pre-training data, randomly sampling 1,000 unseen image-text pairs. The trained model extracted features using average pooling, and a box plot (**Figure in Section G**) visualized the central tendency, spread, and skewness of 1,000 positive and negative pairs. The results show: (i) the median similarity for positive pairs is significantly higher than for negative pairs, indicating clear separation; (ii) while some overlap exists in the interquartile ranges (IQRs), the shift in central tendency confirms the distinction; and (iii) outliers are present, particularly among negative pairs, but they minimally overlap with the core distribution of positive pairs.
>
> - Another example can be found in Table 2 of our paper (we added again below), where we compare two baselines, InfoNCE and PLOT. Both utilize the same contrastive loss, but InfoNCE relies on cosine distance with averaged features, while PLOT directly computes optimal transport over sets of visual and text tokens. The results for these baselines are summarized below.
>
> |                          | VQA-RAD | SLAKE | PathVQA |
> | ------------------------ | ------- | ----- | ------- |
> | InfoNCE (avg.feature)    | 70.34   | 83.29 | 61.6    |
> | PLOT (optimal transport) | 71.89   | 83.16 | 62.4    |
>
> We see that using a more sophisticated distance metric, such as optimal transport, provides a slight improvement (around 1%) over the averaging approach. However, the performance gain is relatively modest. Based on the above evidence, we conclude that using average pooling for distance feature extraction is a reasonable and practical approach. Notably, a similar method was employed in VLAP (Park, Jungin et al., 2024), where its simplicity and consistency were validated.
>
> However, we still acknowledge the Reviewer’s point that alternative methods may offer improvements, and we suggest this as a valuable direction for future exploration.
>
> *Park, Jungin, Jiyoung Lee, and Kwanghoon Sohn. "Bridging Vision and Language Spaces with Assignment Prediction." ICLR 2024*

---

> ### Author Response · Authors · 2024-11-23
> **Response to Reviewer Kirs - Part 2**
>
> **Q3. Compute macro-F1 scores for classification/VQA because medical datasets might be imbalanced.**
>
> Due to time constraints during the rebuttal phase, we focused on the VQA task to report additional macro-F1 scores. These scores were not included in the main paper since other baselines did not provide F1 metrics for VQA, reporting only Recall and Accuracy. To ensure consistency, we computed the F1 scores for both our model and LLaVa-Med using the checkpoints reproduced from their GitHub repository.
>
> We evaluate F1 scores for  VQA-RAD and SLAKE while skipping the PathQA dataset due to its mixed domains, where visual-question answering data cannot be clearly separated by image modalities (e.g., MRI, CT, X-ray) or specific organs (e.g., liver, lung, chest).
>
> - VQA-RAD: questions in this data can be divided into categories of “Chest, Head, Abdomen.” We compute the F1 score for multiple-choice questions for each data body organ. We present the results in **Table 1** in the attached link https://anonymous.4open.science/r/logra-med-386F/readme.md.
>
> - SLAKE: Questions in this data can be divided into different data modalities, such as CT, X-ray, and MRI. We compute the F1 score for multiple-choice questions for each data modality and for all samples. Results are summarized in **Table 2**.
>
> Based on the F1 scores, we conclude that LoGra-Med demonstrates consistent performance, aligning with its accuracy results, and continues to outperform LLaVa-Med on this metric.
>
> **Q4. Confusing about using the “long-context” term.**
>
> We thank the Reviewer for pointing out the potential confusion around the term “long context.” In our work, “long context” does not refer to the length of input tokens used in training LLMs or the window size of attention mechanisms, as is commonly understood in large-scale LLM training. Instead, “long-context” refers to the enriched captions generated by extending the original medical captions with paraphrased descriptions and additional relevant context using GPT-4. These enriched captions provide more detailed semantic information about the input image. **We chose this term to emphasize the richer contextual information contained within the extended captions rather than any technical aspect related to token length or attention span**. To address potential ambiguity, we will revise the terminology in the Abstract, Introduction, and Related Works sections to clearly specify its meaning within the context of our medical visual question-answering task.
>
> Given these arguments, we prefer to retain the current term to minimize significant revisions across multiple sections, including method names and descriptions. However, if the Reviewer feels strongly about this, we are open to reconsidering and adopting an alternative term. Please let us know if this proposed solution is acceptable.
>
> **Q5.Why is graph perspective used? How is it different from optimal transport?**
>
> Our graph alignment solving the node-to-node correspondences under edge constraints indeed can be formulated using optimal transport, namely fused Gromov-wasserstein optimal transport (FGW-OT) (Nguyen, Duy MH, et al. 2024; Ma, Xinyu, et al. 2024). However, **two main challenges hinder us from using optimal transport in LoGra-Med**:
>
> - First, performing the forward pass to compute alignment between graph pairs using optimal transport is computationally expensive (Nguyen, Duy MH, et al. 2024), making it impractical for scaling to large-scale LLM training with hundreds of thousands of samples. This obstacle is further challenged in LoGra-Med, where three separate graph alignment problems have to be solved, significantly incurring computational costs. **In contrast**, adopting a graph-based formulation enables the use of heuristic solvers specifically designed for combinatorial graph matching (Swoboda et al., 2017; Rolínek et al., 2020), providing an efficient solution to address the high computational complexity of graph alignment.
>
> - Second, our training loss (Hamming loss, Eq.(5)) requires gradients from graph alignments to learn feature representations. Using optimal transport would necessitate backpropagation through its Sinkhorn iterations (50-100), adding substantial computational cost and GPU memory usage for storing intermediate variables.
>
> Our graph-based formulation addresses the second challenge by leveraging modern gradient estimation techniques for black-box optimization (Niepert et al. 2021), making the backward step efficient for LLM training.
>
> -------
> *Nguyen, Duy MH, et al. "Structure-Aware E (3)-Invariant Molecular Conformer Aggregation Networks." ICML 2024*
> *Ma, Xinyu, et al. "Fused Gromov-wasserstein graph mixup for graph-level classifications." NeurIPS 2024.*
> *Paul Swoboda et al. “A study of lagrangian decompositions and dual ascent solvers for graph matching.”, CVPR 2017*
> *Niepert et al. “Implicit MLE: backpropagating through discrete exponential family distributions”, NeurIPS 2021*

---

> ### Author Response · Authors · 2024-11-23
> **Response to Reviewer Kirs - Part 3**
>
> **Q6. Without using extended contexts, how LoGra-Med is different from PLOT paper**
>
> We thank the Reviewer for your question. Without using the third graph from the extended caption, LoGra-Med reverts to a graph alignment between the image and its caption. While LoGra-Med and PLOT paper both works with two domain information, there are several significant differences between the two methods that we list below:
> |                                               | LoGra-Med (two graphs)                                                                                                                                                                                                            | PLOT                                                                                                                                                                                             |
> | --------------------------------------------- | --------------------------------------------------------------------------------------------------------------------------------------------------------------------------------------------------------------------------------- | ------------------------------------------------------------------------------------------------------------------------------------------------------------------------------------------------ |
> | Target                                        | Solving a matching between vision and language domains as a part of training multi-modal LLM                                                                                                                                      | Designed for prompt learning where new learnable prompts are added to the frozen text input, propose to use optimal transport distance between visual features and prompt-guided text embedding. |
> | Feature computing                             | Using average pooling of visual/text embeddings                                                                                                                                                                                   | Using local visual features and multiple text embeddings                                                                                                                                         |
> | Distance between a pair of vision-text sample | Using cosine distance given extracted features                                                                                                                                                                                    | Using optimal transport given extracted features                                                                                                                                                 |
> | Training loss                                 | Solving graph alignment between two domains through **combinatorial graph matching**. Compute hamming loss between the output of graph alignment and ground truths. **Backward through black-box gradient estimation techniques.** | **Minimizing pair-wise distance** for each vision-language using OT distance. Backward steps don’t go through optimal mapping.                                                                |
>
> **Q7. Providing some explanation for mixed results with DCI features.**
>
> We aim to integrate DCI into LoGra-Med to capture better multi-scale visual features from vision encoders, potentially benefiting medical image analysis by considering both local (detailed) and global (contextual) features.
>
> To address the Reviewer’s concerns, we retrained LoGra-Med with the DCI component using more carefully optimized hyperparameters. Previously, we had applied the default LoGra-Med settings, including the learning rate, without tailoring them for the DCI-enhanced version. The updated results (**Table 3** in the attached link https://anonymous.4open.science/r/logra-med-386F/readme.md) demonstrate that DCI achieves comparable performance to the standard LoGra-Med on SLAKE, with a slight decrease, while showing improved average performance on VQA-RAD and a particularly notable gain on PathVQA.
>
> We explain this phenomenon because the impact of the DCI component across datasets can be attributed to differences in dataset characteristics and task requirements. In particular, PathVQA likely benefits more from multi-scale features due to its reliance on both local and global visual contexts, making DCI’s integration more effective. On the other hand, VQA-RAD and SLAKE may rely less on multi-scale visual representations and more on different factors like textual reasoning or specific feature interactions. In terms of practicality, we would suggest providing both variations where the better model can be determined based on specific applications.

---

> > ### Author Response · Authors · 2024-11-24
> > **Look forward to your response**
> >
> > Dear Reviewer Kirs,
> >
> > We would like to thank you very much for insightful review, and we hope that our response addresses your previous concerns regarding our paper. However, as the discussion period is expected to end in the next few days, please feel free to let us know if you have any further comments on our work. We would be willing to address any additional concerns from you. Otherwise, we hope that you will consider increasing your rating.
> >
> > Thank you again for spending time on the paper, we really appreciate it!
> >
> > Best regards,
> >
> > Authors

---

> > > ### Comment · Reviewer_Kirs · 2024-11-24
> > > **Reply to Author**
> > >
> > > I really appreciate the thoughtful and comprehensive rebuttal from the authors, especially in comparing and contrasting different methodologies. I have raised my score.

---

> > > > ### Author Response · Authors · 2024-11-24
> > > > **Thank You**
> > > >
> > > > Dear Reviewer Kirs,
> > > >
> > > > We're glad that our rebuttal addresses your concerns and appreciate that you increase your rating to 6.
> > > >
> > > > We will incorporate your suggestions into the revision of our paper as discussed. Please feel free to let us know if you have any further concerns.
> > > >
> > > > Best,
> > > >
> > > > The Authors

---

### Author Response · Authors · 2024-11-23
**General Response to All Reviewers**

We sincerely thank the reviewers for their insightful comments and thorough feedback. The **reviewers found our LoGra-Med algorithm, inspired by the efficiency needs of multi-modal LLM training and implemented through multi-graph alignment, to be innovative** (Reviewer Kirs and Reviewer cEx3). The reviewers also concurred that our use of black-box gradient estimations and efficient heuristic solvers to **address the high time complexity of NP-hard multi-graph matching is well-founded** (Reviewer Kirs and Reviewer cEx3) and supported by robust **theoretical results on metric and geodesic properties** (Reviewer cEx3). Additionally, all reviewers acknowledged the **strength of our comprehensive experiments, spanning three popular VQA datasets, a medical chatbot, and zero-shot classification on 23 datasets**. They also appreciated the well-conducted ablation studies, which demonstrated that (i) LoGra-Med, using only 10% of pre-training data, achieves comparable performance to LLAVA-Med trained with 100% of the data (on VQA tasks), and (ii) outperforms several state-of-the-art multi-modal LLMs.

The reviewers raised concerns, which we addressed in the individual responses below. We summarize the highlights of our responses where additional experiments are provided:

- **Test with simpler solutions, including two graph alignments/using a pair-wise graph alignment solver rather than solving through graph barycenter**:  We provide additional experiments for two settings: (i) performing contrastive learning between two graphs (images and their original captions) and (ii) addressing pairwise graph alignment instead of using graph barycenters in case of three graph alignments. In summary, we observe that (i) achieves lower performance compared to LoGra-Med with three-graph alignment, while (ii) matches the speed of our solver but results in reduced overall performance.


- **Validating using average pooling token features**: We conducted a new experiment by pre-training on 70% of the data and utilizing the trained model to compute distances for positive and negative pairs across 1000 randomly selected unseen samples. The results suggest that the average pooling feature remains an effective approach, as it establishes a clear margin of separation between the two distributions.


- **Evaluating the contribution of long-context vs. multi-graph alignment by pre-training LLava-med baseline with long-context and longer training time:** We ran extra experiments to further validate our approach: (i) comparing the performance of LoGra-Med and the LLaVa-Med baseline, both pre-trained on 40% of the extended context data, and (ii) repeating the same setup as (i), but allowing LLaVa-Med to train for a longer duration than LoGra-Med. The results consistently demonstrate that LoGra-Med outperforms LLaVa-Med in both scenarios by significant margins, underscoring the effectiveness of our multi-graph alignment framework.


- **Validating quality of contexts generated by GPT-4**: We organized a new user study in which four general practitioners, who are our collaborators and are working at a public medical hospital in Vietnam, evaluated the quality of extended captions generated by GPT-4. Given the limited time available during the rebuttal phase, the study was restricted to 1,000 samples, evenly distributed across five data modalities: CT, X-ray, MRI, histology, and others. Till now, we have completed statistical results for three domains, as presented in Figure X, that confirm that the captions generated by GPT-4 are generally consistent with established medical knowledge.


- **Test with different LLM models (simpler and other models) to create extended contexts:**  We present results for LoGra-Med using a more straightforward paraphrase approach (previously included in the ablation studies) alongside new results obtained with the Gemini LLM model for 10% and 40% pre-training scenarios. The performance of the models on two VQA tasks demonstrates that LoGra-Med maintains its effectiveness even when integrated with the Gemini LLM model.

- **Others: clarifying setting of 10% pre-training data, number of K-nearest neighbors in graph construction steps, and interpolating between auto-regressive and graph-alignment in the pre-training phase:**  We provide detailed clarifications regarding the pre-training data in the response below and conducted four additional experiments: (i) varying the number of neighbors in graph construction using KNN with K = 3 and K = 7, and (ii) exploring the trade-off between auto-regressive modeling and multi-graph alignment using different coefficient values. The outcome results validate our choices on K = 5 and coefficient = 1.

We summarize new experiments ran during the rebuttal phase in this link:

https://anonymous.4open.science/r/logra-med-386F/readme.md

---

### Note · Authors · 2025-01-23

I have read and agree with the venue's withdrawal policy on behalf of myself and my co-authors.